



# Comparing the Impacts of Single- and Multi-Objective Optimization on the Parameter Estimation and Performance of a Land Surface Model

Cong Xu[1], Gaofeng Zhu[1], Yang Zhang[1], Kun Zhang[2]

[1]College of Earth and Environmental sciences, Lanzhou University, Lanzhou 730000, China
[2]School of Biological Sciences and Institute for Climate and Carbon Neutrality, the University of Hong Kong, Hong Kong SAR, China

*Corresponding author:* Gaofeng Zhu (zhugf@lzu.edu.cn)



**Abstract**

In land surface models (LSMs), precise parameter specification is crucial to reduce the inherent uncertainties and
enhance the accuracy of the simulation. However, due to the multi-output nature of LSMs, the impact of different
optimization strategies (e.g., single- and multi-objective optimization) on the optimization outcome and efficiency
remains ambiguous. In this study, we applied a revised particle evolution Metropolis sequential Monte Carlo (PEM-
SMC) algorithm for both single- and multi-objective optimization of the Common Land Model (CoLM), constrained
by latent heat flux (*LE*) and net ecosystem exchange (*NEE*) measurements from a typical evergreen needle-leaf
forest observation site. The results reveal that the revised PEM-SMC algorithm, demonstrates a robust ability to
tackle the multi-dimensional parameter optimization challenge for LSMs. The sensitive parameters for different
target outputs can exhibit conflicting optimal values, resulting in single-objective optimization improving the
simulation performance for a specific objective at the expense of sacrificing the accuracy for other objectives. For
instance, solely optimizing for LE reduced the root-mean-square error (RMSE) of the simulated and observed LE by
20% but increased the RMSE of the NEE by 97%. Conversely, multi-objective optimization can not only ensure that
the optimized parameter values are physically sound but also balances the simulation performance for both LE and
NEE,  as evidenced by the decrease in RMSE for LE and NEE of 7.2 W/m2 and 0.19 μmol m-2 s-1, respectively. In
conclusion, these findings reveal that comprehensively integrating the various available observational data for multi-
objective optimization is preferable for parameter calibration in complex models.

**1 Introduction**

Global warming has led to increasingly intricate dynamics between terrestrial surfaces and the atmospheric system,
necessitating advancements in the precision of the representations utilized within models, particularly those related to
terrestrial surface processes. Land surface models (LSMs) play a crucial role in accurately simulating the exchange of
water, carbon, and energy between the land surface and atmosphere, understanding biosphere-climate interactions,
and evaluating global climate change impacts (Field et al., 2004; Henderson-Sellers et al., 1995; McGuire et al., 2001).
However, LSMs are challenged by their complexity and the difficulty in accurately determining the numerous inherent
parameters, leading to significant uncertainties in simulating the land-atmosphere flux exchanges at a large scale
(Folberth et al., 2019; Li et al., 2020; Luo et al., 2016; Thum et al., 2017). Therefore, parameter calibration/estimation
which integrates multi-source observational data to optimize the model parameters has emerged as a fundamental step
in diminishing these uncertainties and enhancing the efficacy of the models in simulating the interaction processes
between land and atmosphere under the current and future exacerbated climate change conditions (Duan et al., 2017).

LSMs generally encompass multiple interdependent processes, where the configuration of the parameters has a
concurrent impact on the accuracy of the diverse output simulations (Bastrikov et al., 2018). The traditional parameter
calibration methods for LSMs predominantly target single-objective optimization, emphasizing parameter adjustment
to enhance the simulation performance for specific processes (e.g., surface carbon and water fluxes) (Kato et al., 2013;
Li et al., 2018; Ricciuto et al., 2011; Sellers et al., 1989; Xia et al., 2004b). However, given the intricate nature of
LSMs and the interactions among the multiple outputs, optimization targeting a single output can inadvertently
compromise the simulation accuracy for the other output variables. In recent years, multi-objective optimization
algorithms have gained increasing attention for their capacity to provide a balanced solution for multiple conflicting
objectives (Bastidas et al., 1999; Saini et al., 2021; Segura et al., 2016). These algorithms adeptly navigate the dynamic
input-output competition in complex models, achieving an optimal balance in overall model performance.
Consequently, multi-objective optimization is increasingly favored for dealing with the parameter estimation of



complex models with numerous outputs and interactive processes, such as hydrological models (Gupta et al., 1999;
Vrugt et al., 2003b), LSMs (Gong et al., 2015; Leplastrier et al., 2002; Varejao et al., 2013), and soil-vegetation-
atmosphere coupled models (Liu et al., 2005; Pollacco et al., 2013). Despite this, most of the existing research
primarily focuses on the development and implementation of various multi-objective algorithms for parameter
estimation, with limited studies offering comprehensive scientific substantiation and practical validation for the
superiority of the multi-objective optimization strategy over single-objective optimization. Critical considerations
include the potential detriment of single-objective optimization to non-target outputs and whether multi-objective
optimization can concurrently improve the accuracy for multiple outputs. Furthermore, the distinctions between
single- and multi-objective optimization in terms of parameter estimation, model performance improvement, and
application reliability remain to be clarified.
The introduction of expanded parameter spaces and increased optimization complexity by multi-objective
optimization emphasizes the importance of developing more efficient global optimization algorithms. Over the past
few decades, numerous optimization algorithms have been employed in the LSMs to obtain appropriate parameter
values, including genetic algorithms (D'heygere et al., 2006; Ines et al., 2008), particle swarm optimization (Eberhart
et al., 2001; Gill et al., 2006; Zhang et al., 2009), shuffled complex evolution (SCE) (Duan et al., 1993, 1994), the
Markov chain Monte Carlo (MCMC) algorithm (Smith et al., 2008; Van et al., 2005; Zhang et al., 2019), and the
sequential Monte Carlo (SMC) algorithm (Dong et al., 2023; Jeremiah et al., 2011; Zhu et al., 2018). Among these
algorithms, the SMC samplers, which are also known as particle filters, are recognized for theoretically providing a
direct and effective way of estimating the posterior distribution through a series of gradual approximations and weight
redistributions (Doucet et al., 2001; Fan et al., 2008). However, the traditional SMC samplers face the challenge of
the particle impoverishment problem, which is a consequence of the resampling step that discards less significant
particles in favor of duplicating more promising ones. To combat this, candidate particle generation algorithms using
an MCMC transition kernel have been implemented in the SMC moving step to enhance the particle diversity and
quality [e.g., the random walk Metropolis (RWM) algorithm (Metropolis et al., 1953) and the adaptive random walk
Metropolis (ARM) algorithm (Chopin, 2002; Jeremiah et al., 2011)]. In line with these advancements, our previous
research introduced the particle evolution Metropolis (PEM) method, which is a novel candidate-generating approach
that combines the appealing aspects of genetic and evolutionary algorithms with the robustness of the Metropolis-
Hasting (M-H) algorithm (Zhu et al., 2018). In the case study on a synthetic multi-dimensional bimodal normal
distribution, the PEM-SMC sampler demonstrated a superior efficiency in exploring high-dimensional and complex
parameter spaces, compared to other SMC samplers (i.e., RWM-SMC and ARM-SMC). Nevertheless, while the
genetic-styled operations (e.g., crossover and mutation) enhance the particle diversity, they simultaneously impose
significant computational burdens, necessitating some redundant calculations in the original algorithm for each newly
generated particle. This limitation is particularly pronounced in the parameter optimization of complex LSMs with
extensive single-run durations. In this paper, based on our previous work, we present a revised PEM-SMC algorithm
that further refines the particle candidate mechanism in the moving step to improve the computational efficiency while
maintaining the optimization efficiency. Moreover, while our previous research focused on the performance of the





PEM-SMC sampler in simple hydrological model parameter optimization, the current paper explores its applicability
and potential in the high-dimensional parameter optimization of more complex LSMs.

This paper provides a comprehensive parameter sensitivity analysis and optimization of the Common Land Model
(CoLM) based on the latent heat flux (*LE*) and net ecosystem exchange (*NEE*) measurements from a FLUXNET
observation site and extensively evaluates the impact of different optimization strategies on the model simulation
performance. The study encompasses: (a) the application of both qualitative and quantitative sensitivity analysis
methods to precisely identify the pivotal parameters for accurately simulating water and carbon processes in the CoLM;
(b) a validation of the efficacy of the integrated optimization framework combining sensitivity analysis with the
modified PEM-SMC algorithm in optimizing the multi-dimensional parameters of complex LSMs; and (c) the
implementation of both single-objective and multi-objective optimization of the model parameters to elucidate the
distinctions in the optimization outcome and efficiency attributed to different constraints. The novel optimization
algorithm proposed in this paper, coupled with the extensive investigation into different optimization strategies,
provides methodological insights for the parameter optimization of LSMs.
**2 Materials and Methods**
**2.1 The CoLM and Adjustable Parameters**
The Common Land Model (CoLM), as developed by Dai et al. (2003), has significantly evolved from its initial form
into a globally acclaimed third-generation LSM. The CoLM is characterized by its intricate, comprehensive, and
precise representation of biophysical, biochemical, ecological, and hydrological processes on the land surface, and
has been widely used in the simulation of energy, momentum, water, and carbon transport between the land and
atmosphere (Ment et al., 2009; Zeng et al., 2002).

In this study, we strategically selected 40 out of the 46 time-invariant parameters from the CoLM, deliberately
excluding certain of the model's internal parameters, to conduct a comprehensive sensitivity analysis and optimization
of the model parameters (Table S1). These chosen parameters, which are inherently static, represent the physical
properties of vegetation and soil and can be adjusted according to the specific local environmental conditions. For
ease of reference, these parameters were indexed from P1 to P40. The predefined initial range of these parameters
significantly influences the results of the high-dimensional parameter sensitivity analysis and optimization. Ensuring
the objectivity and rationality of the parameter range is crucial for the validity of the final calibration results.
Consequently, the range for the 40 parameters was established based on a literature review, the local environmental
conditions at the study site, and the biophysical/chemical meaning of each specific parameter (Sellers et al., 1996; Ji
et al., 2010; Li et al., 2013). A detailed description of this process is provided in Sect. S2.



## 2.2 Design of the Parameter Sensitivity Analysis

Prior to the optimization, a global sensitivity analysis of the parameters was conducted to assess the parameter importance via a relatively inexpensive "coarse" sampling of the parameter space. Among the sensitivity analysis techniques, qualitative methods enable the identification of crucial parameters using a relatively small sample size (hundreds to thousands), albeit with significant outcome variability among different methodologies. In contrast, quantitative methods based on variance decomposition provide superior precision but require extensive datasets (from tens to hundreds of thousands). To navigate these challenges, we identified the 10 most sensitive parameters by integrating the results from three qualitative methods and subsequently ranked them by employing a quantitative method for further refinement (Li et al., 2013). Here, we provide a brief description of how these methods are applied in variable selection.

(a) The delta test (DT) method: a noise variance estimator based on the concept of nearest neighbors (NNs) (Eirola et al., 2008; Pi et al., 1994). For a given set of input parameters $\theta_i (i = 1, \dots, m)$ and associated output $Y$, the assumption is that there is a functional dependence between them:

$$Y = f(\theta_i) + \varepsilon_i \tag{1}$$

where $\varepsilon_i$ is an independent identically distributed random variable with zero mean. Noise variance estimation is the study of how to give an "a priori estimate" for $\delta(\varepsilon)$. The NN of a point is defined as the unique point that minimizes a Euclidean distance to that point in the input space:

$$N(i) := \arg\min_{j \neq i} \|\theta_i - \theta_j\|^2 \tag{2}$$

The DT criterion of a variable subset $s \subseteq \{\theta_1, \dots, \theta_m\}$ is then written as:

$$\delta(s) = \frac{1}{2N} \sum_{i=1}^{N} \left(Y_{N_s(i)} - Y_i\right)^2 \tag{3}$$

where N is the sample size, $Y_i$ is the function value corresponding to $\theta_i$, and $Y_{N_s(i)}$ is the function value corresponding to the NNs of the input point $\theta_i$ for subset $s$. Consequently, the variable subset $s$ with the smallest DT criterion is the most sensitive parameter.

(b) The multivariate adaptive regression splines (MARS) method: a non-parametric regression technique (Friedman, 1991) that employs a specific class of basis functions as predictors, replacing the original input variables. The general form of the MARS model can be expressed as:

$$Y = \beta_0 + \sum_{j=1}^{M} \beta_j B_j(\theta_i) \tag{4}$$

where $\theta_i (i = 1, \dots, m)$ is the vector of the inputs; $B_j$ is the $j$-th basis function, which can be a single spline function or a product of two or more basis functions; and the coefficients $\beta_j s$ are estimated by minimizing the sum of squared residuals (Shahsavani et al., 2010). In fact, the MARS regression model is constructed by fitting these basis functions to various intervals of the independent variables. The final model in MARS is developed through a forward-backward procedure: initially, an over-fitted model is constructed by considering all the variables in the forward step; subsequently, this model is pruned by sequentially eliminating variables in the backward step, thereby creating a new model G. The performance of each model G is evaluated using generalized cross-validation (GCV):



$$GCV(G) = \frac{1}{N} \frac{\sum_{i=1}^{N}(O_i - Y_i)^2}{\left[1 - \frac{C(G)}{N}\right]^2} \tag{5}$$

where $N$ is the number of samplers; $O_i$ and $Y_i$ are the i-th observed and estimated values, respectively; $C(G)$ is the number of effective parameters, and is equal to $1 + c(G)d$; $d$ is the effective degrees of freedom; and $c(G)$ is a penalty for adding a basis function. The increase in GCV values between the pruned and over-fitted models is employed as a metric to gauge the importance of the eliminated variables: a larger increase in GCV values signifies greater importance of the removed variable (i.e., a sensitive parameter).

(c) The Morris method: a gradient-based sensitivity analysis technique using an individually randomized Morris one-factor-at-a-time (MOAT) design (Campolongo et al., 2007; Morris, 1991). It involves calculating multiple incremental ratios, termed elementary effects, for each input variable (parameter) and averaging these effects to assess the overall importance of the input variables. Campolongo et al. (2007) introduced a refined version of the elementary effects method. In this approach, the model parameters $\theta_i (i = 1, \ldots, m)$ are assumed to vary across $p$ specified levels within the input factor space, creating an experimental region $\Omega$ that constitutes an m-dimensional p-level grid. For a given input $\theta^0 = (\theta_1, \theta_2, \ldots, \theta_m)$, the elementary effect of variable $\theta_j$ is defined as:

$$d_j = \frac{f(\theta_1, \ldots, \theta_j + \Delta, \ldots, \theta_m) - f(\theta_1, \ldots, \theta_j, \ldots, \theta_m)}{\Delta} \tag{6}$$

where $\Delta$ is a value in $1/p - 1, \ldots, p - 2/p - 1$. The sampling strategy entails randomly determining the starting point of each trajectory and perturbing each input variable by either $+\Delta$ or $-\Delta$ in random order. At the end of the process, a trajectory spanning m+1 points is evaluated to compute the elementary effects for all m input variables. The mean ($\mu_j$) and standard deviation ($\sigma_j$) of the elementary effects ($d_j$) serve as indicators of the sensitivity of the input variable $\theta_j$:

$$\mu_j = \sum_{i=1}^{r} |d_j(i)|/r \tag{7}$$

$$\sigma_j = \sqrt{\sum_{i=1}^{r} \left(d_j(i) - \frac{\sum_{i=1}^{r} d_j(i)}{r}\right)^2 / r} \tag{8}$$

where $\mu_j$ assesses the overall influence of $\theta_j$ on the output, while $\sigma_j$ estimates the higher-order effects (i.e., effects due to interactions) of $\theta_j$.

(d) The Sobol' method: a quantitative sensitivity analysis approach based on variance decomposition (Sobol', 1993). It decomposes the total variance of outputs Y into a summation of incrementally dimensional terms:

$$V = \sum_{i=1}^{m} V_i + \sum_{i=1}^{m-1} \sum_{j=i+1}^{m} (V_{ij} + \cdots + V_{1,\cdots,m}) \tag{9}$$

where $m$ is the number of input variables (parameters), $V_i$ represents the part of the output variance attributable to the individual input parameter $\theta_i (i = 1, \ldots, m)$ (first-order sensitivity), $V_{ij}$ represents the part of the output variance resulting from the interaction between input variables ($\theta_i$ and $\theta_j, i \neq j$) (second-order sensitivity), and $V_{1,\cdots,m}$ represents the part of the output variance which can be explained by the interaction of all the variables. In this paper, the total effect of $\theta_i$ is utilized as the metric for assessing its sensitivity, computed by:

$$S_i = 1 - \frac{V_{-i}}{V} \tag{10}$$

where $V_{-i}$ represents the variance computed excluding variable $\theta_i$. The Sobol' method rigorously quantifies the relative contribution of the individual parameters and their interactive effects on the total variance in the model output, necessitating an extensive sample dataset (104 to 105 or more).






The specific processes of the sensitivity analysis for the three targets (*LE/NEE/LE+NEE*) are detailed as follows: (a)
Sample generation: an ensemble of 400 samples within the prior range of 40 parameters was generated for the DT and
MARS methods using the Latin Hypercube (LH) sampling method (Deutsch et al., 2012). For the MOAT method, a
distinct ensemble of 410 (10 multiples of n+1, where n is the number of parameters) was created, utilizing the Monte
Carlo (MC) sampling approach (Hastings, 1970). (b) Cost function calculation: Both the 400 and 410 ensembles were
used to drive the CoLM, followed by computing the cost function values to measure the discrepancy between the
simulations and observations. Given the varying magnitudes of the target variables (LE/NEE), we employed the
normalized root-mean-square error (NRMSE) as the cost function:
$$NRMSE = \frac{\sqrt{\sum_{t=1}^{T} S(t) - O(t))^2}}{\sum_{t=1}^{T} O(t)} \tag{11}$$

where $T$ is the total number of simulations; and $S(t)$ and $O(t)$ are the simulated and observed values of the target
variables, respectively. For multi-objective sensitivity analysis (*LE+NEE*), a weighting function-based method was
utilized to transform the multiple objectives into a single objective:
$$F = \sum_{i=1}^{m} NRMSE_i \tag{12}$$

where $i$ is the index of the target variables, and $m$ denotes the number of objectives. (c) Sensitivity analysis: The input
($\theta$)-output ($F$) sample pairs were analyzed using four qualitative methods to discern the sensitivity of each parameter
to the target outputs. (d) Parameter selection: The mean sensitivity for each parameter was calculated across the four
qualitative analysis outcomes, leading to the selection of 10 parameters with the highest sensitivity for further
quantitative analysis. (e) Sample regeneration and cost function re-calculation: A new set of 100,000 samples for the
10 parameters was generated via LH sampling, followed by a repetition of step 2 to calculate the corresponding cost
function value. (f) Parameter determination for optimization: The total sensitivity results, derived from the Sobol'
method applied to the 100,000 input-output sample pairs, guided the determination of parameters for the subsequent
optimization. All the sensitivity analyses were conducted using the Problem Solving Environment for Uncertainty
Analysis and Design Exploration (PSUADE) software package (Tong, 2005) at the Supercomputing Center of
Lanzhou University in China.
**2.3 Parameter Optimization with the Revised PEM-SMC Algorithm**
Within the Bayesian single-objective optimization framework, the parameters are conceptualized as probabilistic
variables, with the posterior parameter distribution formulated as:
$$p(\theta \mid D) \propto p(\theta)p(D \mid \theta) \tag{13}$$

where $D = \{O_{1:T}\}$ is the set of observations of the target variable; $T$ is the total number of observed data; $\theta$ represents
the parameters; $p(\theta)$ and $p(\theta \mid D)$ respectively denote the prior and posterior distributions of the parameters;
and $p(D \mid \theta)$ represents the model likelihood, which can be expressed as follows (Zhu et al., 2014):
$$p(D \mid \theta) = (2\pi\sigma^2)^{-T/2} \prod_{t=1}^{T} \exp\left\{-\frac{[O(t) - S(X_t; \theta)]^2}{2\sigma^2}\right\} \tag{14}$$

where $O(t)$ and $S(X_t; \theta)$ denote the observed and simulated sequences of the target variable at each time step
t(t=1,2,…, T), respectively. The latter is driven by the forcing data $X_t$ and parameters $\theta$. $\pi$ is a mathematical constant,





and $\sigma$ denotes the standard deviation of the measurement error, which can be estimated using the analytical method
(Braswell et al., 2005):

$$\sigma = \sqrt{\frac{1}{T}\sum_{t=1}^{T}[O(t) - S(X_t; \theta)]^2} \tag{15}$$

For multi-objective optimization, the posterior parameter distributions are expressed as the product of the prior
distribution and multiple likelihood functions, paralleling the approach in the weighting function-based methods:

$$p(\theta \mid D) \propto p(\theta) \prod_{i=1}^{m} \prod_{ti=1}^{T_i} p(O_{ti}^i | \theta) \tag{16}$$

where $m$ is the number of objective variables, and D= $\{O_{1:T_1}^1, O_{1:T_2}^2, \dots, O_{1:T_m}^m\}$ denotes the observation sets of the i-
th(i=1,2,…, m) objective. $p(O_t^i | \theta)$ represents the model likelihood for objective $i$ and is calculated by Equations 14
and 15.

Owing to the unfeasibility of deriving a direct analytical solution for the integral in Equations 14 and 16, the SMC
sampler is utilized to generate a sequence of weighted particles, thereby approximating the posterior distribution of
parameters $p(\theta \mid D)$. However, since it is difficult to sample directly from $p(\theta \mid D)$, the SMC sampler alternatively
samples from a sequence of intermediary distributions $\pi_s(\theta)$ constructed by the geometric bridge method (Del Moral
et al., 2006):

$$\pi_s(\theta) \propto p_0(\theta)^{1-\beta_s} p(\theta \mid D)^{\beta_s} \tag{17}$$

where $p_0(\theta)$ and $\pi_s(\theta)$ denote the initial and the s-th distribution in the sequence (s=0, 1, …, S), respectively. $\beta_s$ is a
sequence of scalar powers, such that $0 \le \beta_0 \le \beta_1 \le \cdots \le \beta_S = 1$, which allows a gradual transition of $\pi_s(\theta)$ from
the initial distribution $\pi_0(\theta) \propto p_0(\theta)$ when $\beta_0 = 0$ to the posterior distribution $\pi_S(\theta) \propto p(\theta \mid D)$ when $\beta_S = 1$.
Following Jeremiah et al. (2011, 2012), an exponential ($\beta_s$) sequence is used in the PEM-SMC method.

Upon determining the number of particles ($Np$) and the number of evolutions ($S$), the SMC sampler employs a series
of steps—reweighting, resampling, and moving—to transition particles from distribution $\pi_{s-1}(\theta)$ to $\pi_s(\theta)$. In the
reweighting step, particles more closely aligned with the posterior distribution $\pi_s(\theta)$ are assigned greater weights $w_j^s$,
enhancing their influence in the samplers. The subsequent resampling step addresses the issue of less significant
particles, termed "bad" particles. These are replaced with exact replicas of more promising particles through the
systematic resampling method. This step plays a pivotal role in ensuring the algorithm's convergence, facilitating a
gradual transition of particles from the prior to the posterior distribution. However, a notable challenge arises from
the resampling step: the potential reduction in particle diversity, leading to insufficient exploration of the parameter
space. To mitigate this, in our previous study (Zhu et al., 2018), we introduced a new candidate-generating method
named particle evolution Metropolis (PEM). This method integrates genetic algorithm features—crossover and
mutation operators—into the M-H algorithm framework. In the crossover operator, each parental chromosome pair
$\theta_i^s$ and $\theta_j^s$ ($i \ne j, i = 1,2, \dots, N/2, s = 1,2, \dots, S$) is selected to create a new offspring pair $\bar{\theta}_i^s$ and $\bar{\theta}_j^s$ using the one-
point crossover operator. The new offspring pair is accepted with probability $\min\left\{1, \frac{\pi_s(\bar{\theta}_i^s)\pi_s(\bar{\theta}_j^s)}{\pi_s(\theta_i^s)\pi_s(\theta_j^s)}\right\}$ according to the





M-H rule; otherwise, the current parental pair remains unchanged. In the mutation operator, each chromosome $\theta_i^s (i = $
$1,2, \dots, N)$ is used to create a new chromosome $\bar{\theta}_i^s$ according to the differential evolutionary algorithm:

$$\bar{\theta}_i^s = \theta_i^s + \gamma\left(\theta_{r_1}^s - \theta_{r_2}^s\right) + \zeta_d \tag{18}$$

where $r_1$ and $r_2$ are integer values without replacement from $\{1, \dots, j - 1, j + 1, \dots, N\}$; $\gamma = 2.38/\sqrt{2d}$ denotes the
jump rate; and $\zeta_d \sim N_d(0, b^*)$ is drawn from a normal density with a small standard deviation, say $b^* = 10^{-6}$. The
new chromosome $\bar{\theta}_i^s$ is accepted with the probability $\min\left\{1, \frac{\pi_s(\bar{\theta}_i^s)}{\pi_S(\theta_j^s)}\right\}$.

While the integration of crossover and mutation operators in the SMC algorithm enriches the particle diversity, it also
imposes significant computational burdens. To improve the algorithm's efficiency, we made pivotal modifications to
the original PEM-SMC algorithm, which include: (a) Elimination of the crossover operator: In the revised algorithm,
the moving step exclusively employs the mutation operator to generate new particles, abandoning the crossover
operator. This adjustment addresses the inefficiency inherent in the crossover operator, which recombines parameters
without value alteration. This often results in the time-intensive generation of particles with low acceptance probability
under the M-H rule and a risk of encountering the unexplained hyperparameter problem. (b) Modification of the
mutation operator execution conditions: Unlike the original PEM-SMC algorithm, where the mutation is conditional
upon the effective sample size $N_{eff} = 1/\sum_{i=1}^{N}\left(w_j^s\right)^2$ being less than half the number of particles, the mutation operator
is now uniformly employed at every evolutionary step for each particle. This strategic alteration promotes greater
particle diversity and prevents stagnation at local optima, albeit at the cost of some computational efficiency. (c)
Runtime reduction: Through comparative analysis and validation, including a synthetic five-dimensional bimodal
normal distribution and a benchmark experiment on the CoLM, it is evident that the revised PEM-SMC algorithm
maintains its optimization efficacy while significantly reducing the runtime by over 40% (see Sect. S1). This time
reduction is crucial, especially for parameter optimization in complex models with lengthy single-run durations.
Overall, these modifications substantially enhance the PEM-SMC algorithm, striking a balance between efficiency
and a thorough exploration of the parameter space. The pseudo-code of the revised PEM-SMC algorithm is detailed
below:
*STEP 1: Initialization*
(a) Draw an initial population $\left\{\theta_j^0\right\}(j = 1,2, \dots, N)$ from the prior distribution $p_0(\theta)$, and set weights $w_i^0 = 1/N$.
(b) Determine the exponential sequence $0 \le \beta_0 \le \beta_1 \le \cdots \le \beta_S = 1$.
*FOR $s \leftarrow 1, 2, \dots, S$ DO (stage evolution)*
*STEP 2: Reweighting*
(a) Set $\theta_j^s = \theta_j^{s-1} (j = 1, \dots, N)$, and calculate the weight $w_j^S = w_j^{S-1}\frac{\pi_s(\theta_j^S)}{\pi_{s-1}(\theta_j^{S-1})}$.
(b) Normalize the weight so that $\sum_{j=1}^{N} w_j^S = 1$.
*STEP 3: Resampling*





(a) Calculate the effective sample size $N_{eff}$

(b) if $N_{eff} < N/2$, resample from $\{\theta_i^s, w_i^s\}(j = 1,2, \ldots, N)$ based on the systematic resample procedure and

set $w_j^s = 1/N(j = 1,2, \ldots, N)$; otherwise, go to the next step.

*STEP 4: Mutation*

*FOR $j \leftarrow 1,2, \ldots, N$ DO (mutation operator)*

(a) For each chromosome $\theta_j^s$, create a new chromosome $\bar{\theta}_j^s$ by

(b) with probability $\min\left\{1, \dfrac{\pi_s(\bar{\theta}_i^s)\pi_s(\bar{\theta}_j^s)}{\pi_s(\theta_i^s)\pi_s(\theta_j^s)}\right\}$, set $\theta_i^s = \bar{\theta}_i^s$ and $\theta_j^s = \bar{\theta}_j^s$, else leave $\theta_i^s$ and $\theta_j^s$ unchanged.

*END FOR (mutation operator)*

*END FOR (stage evolution)*

After determining the parameters ($\theta$) used for optimization through the sensitivity analysis, we employed the revised
PEM-SMC algorithm to automatically calibrate the selected parameters. The two control variables in the PEM-SMC
sampler, i.e., the number of particles in the population $Np$ and the number of evolutions $S$, were set to 200 and 100,
respectively. The parameters of the CoLM were optimized by the observed LE and NEE separately and simultaneously.
For ease of description, the single-objective and multi-objective simultaneous optimizations constrained by the LE
and NEE fluxes are denoted as Opt_LE, Opt_NEE, and Opt_ALL, respectively. The revised PEM-SMC algorithm,
written in MATLAB, was deployed for the parameter optimization of the CoLM at the Supercomputing Center of
Lanzhou University.
**2.4 Study Site and Model Performance Evaluation**
Encompassing approximately one-third of the Earth's forests, the boreal forest ecosystem constitutes the most
substantial terrestrial biomass reservoir, significantly influencing global climate regulation. The precise modeling of
the intricate hydrocarbon dynamics within these ecosystems is pivotal for advancing our understanding of global
terrestrial carbon storage and climate dynamics (Pan et al., 2011; Bradshaw et al., 2015). In this study, we focused on
RU-FY2, which is a typical evergreen needle-leaf forest (ENF) observation station, strategically situated in the Central
Forest Reserve of the Tver region of Russia. Positioned at 32°54′E, 56°27′N, this site experiences a warm humid
continental climate with an average annual temperature of 4.39 ℃ and precipitation of 668.53 mm. The predominant
vegetation is dry spruce. Data for this site, encompassing conventional meteorological and eddy covariance
measurements, was sourced from the FLUXNET community. This meteorological dataset, spanning from 2015 to
2020 and characterized by high quality with no missing entries, includes half-hourly variables such as downward
short-wave and long-wave radiation, precipitation, specific humidity, temperature, atmospheric pressure, and wind
speed in eastward and northward directions. The multi-year average energy closure rate of the flux observations from
2015 to 2020 was near 100% (Fig. S6), signifying the exceptional quality of the energy flux observations. For *NEE*,
we selected the mean values of two variables— NEE_CUT_MEAN and NEE_VUT_MEAN—as the observational





metrics. Given the minimal nature of water and carbon fluxes in the non-growing season, coupled with their uncertain
influence on parameter optimization effectiveness, our analysis was exclusively concentrated on the modeling of these
fluxes during the growing season. The meteorological driver data from 1 June 2015 to 31 August 2019 were repeated
10 times to spin up the CoLM, while model simulations from 1 June 2020 to 31 August 2020 with a half-hour time
step were used for the model parameter sensitivity evaluation and optimization.

We quantified the difference between the simulated and observed target variables (*LE* and *NEE*) in the different
optimization scenarios using the root-mean-square error (RMSE), the Nash-Sutcliffe efficiency coefficient (NSE), and
the Pearson's correlation coefficient (R):
$$RMSE = \sqrt{\frac{1}{T}\sum_{t=1}^{T}(S(t) - O(t))^2} \qquad (19)$$

$$NSE = 1 - \frac{\sum_{t=1}^{T}(S(t) - O(t))^2}{\sum_{t=1}^{T}[O(t) - \bar{O}(t)]^2} \qquad (20)$$

where $O(t)$ and $S(t)$ are respectively the observed and simulated values for each simulation point *t(t=1,2,..., T)*, and
$\bar{O}(t)$ is the mean of the observed data. The NSE serves to quantitatively assess the precision of the model outputs in
relation to the observed data, with a range from $-\infty$ to 1. The closer the NSE is to 1, the more accurate the simulation
is.
**3 Results**
**3.1 Parameter Sensitivity**
The results of the qualitative parameter sensitivity analysis among the different methods are shown in Fig. 1. For every
target variable, the different sensitivity analysis methods obtained similar results in screening out the sensitive
parameters, although there are some discrepancies in the sensitivity scores. For example, in the case of *Opt_LE*, all
the methods identified the same three most sensitive parameters—P33, P34, and P35—while the MOAT method
screened out more moderately sensitive parameters (e.g., P2, P7, P13, P18, etc.). Therefore, it is most reasonable to
identify the sensitive parameters according to the sum (or mean) of the sensitivity values obtained by the different
qualitative analysis methods. In the three optimization scenarios, there are some common most sensitive parameters
(e.g., P33, P34, P35, P3, etc.) and some individually sensitive parameters. For example, P8, P7, P36, and P13 are
sensitive to *Opt_LE*, while they are not the sensitive parameters of *Opt_NEE* and *Opt_ALL*. Meanwhile, we can see
that the sensitive parameters of *Opt_NEE* and *Opt_ALL* are essentially the same, which could indicate that the *NEE*
observations are more restrictive on the parameters than the *LE* observations at this site. Based on the results of the
qualitative analysis, we selected the 10 parameters with the highest sensitivity scores for each target variable to
perform the subsequent Sobol' quantitative sensitivity analysis: *Opt_LE* (P34, P33, P35, P8, P3, P7, P36, P18, P13,
P32); *Opt_NEE* (P34, P33, P3, P35, P9, P5, P30, P29, P31, P18); and *Opt_ALL* (P34, P33, P3, P40, P5, P9, P35, P37,
P29, P6).

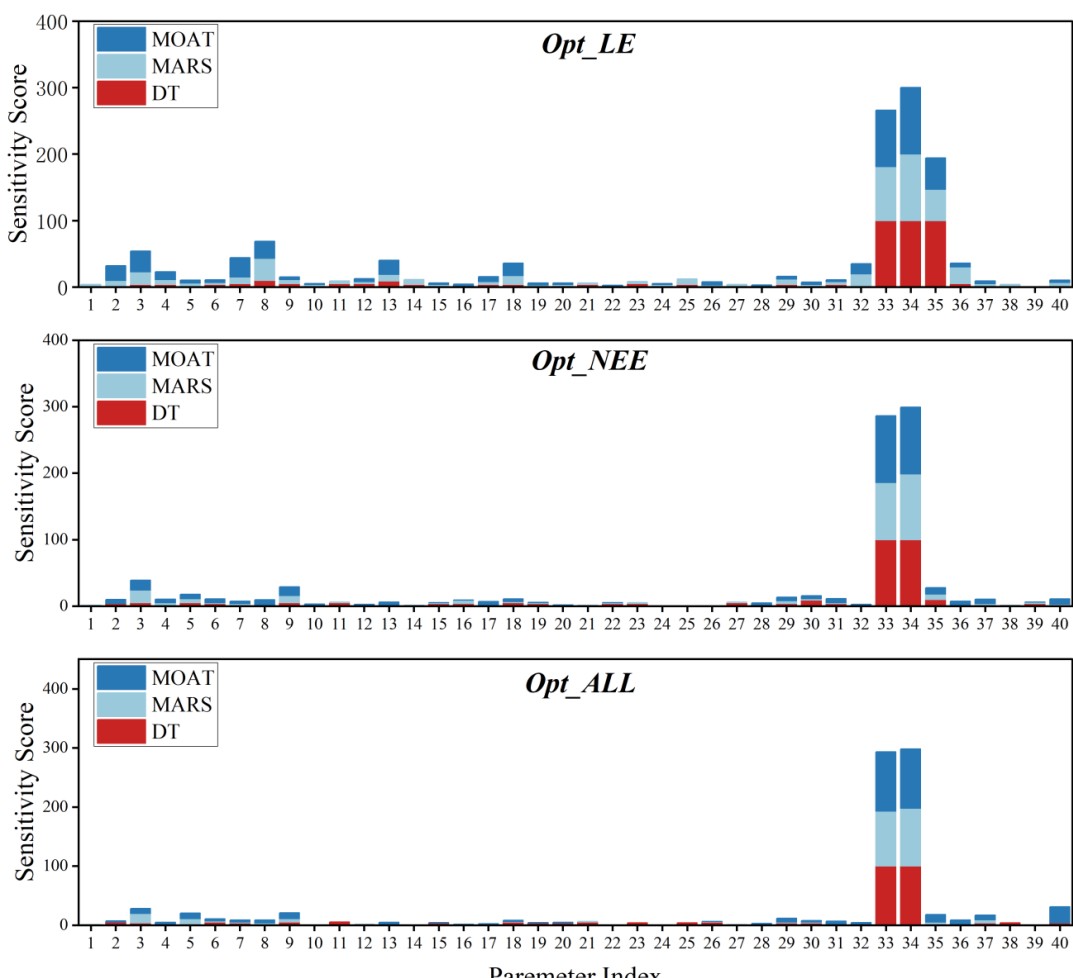

**Figure 1 The sensitivity scores of the 40 parameters to different target variables (*Opt_LE*, *Opt_NEE*, and *Opt_ALL*) based on the different sensitivity analysis methods (MOAT, MARS, and DT). The sensitivity score is the sum of the results from the three methods.**

The screening results obtained by the four qualitative methods indicated that P34 and P33 are the most sensitive parameters for the three target variables, which is consistent with the Sobol' quantitative analysis results, as shown in Fig. 2. The Sobol' results showed that P34 and P33 can explain 65%, 93%, and 77% of the total variance between the simulated and observed values of the three target variables, respectively. Based on the principle that the cumulative relative importance of the parameters is greater than 95% (which means that the variance can essentially be explained by these parameters), we selected the most sensitive parameters for each target variable to perform the subsequent optimization: *Opt_LE* (P34, P33, P35, P8, P36, P3); *Opt_NEE* (P34, P33, P35, P30, P3); and *Opt_ALL* (P34, P33, P35, P9, P5, P37). Since the selected parameters can explain more than 95% of the total variance of the model output, we





believe that taking them as the optimized parameters instead of all 40 parameters to calibrate the model can improve
the optimization efficiency without losing effectiveness.

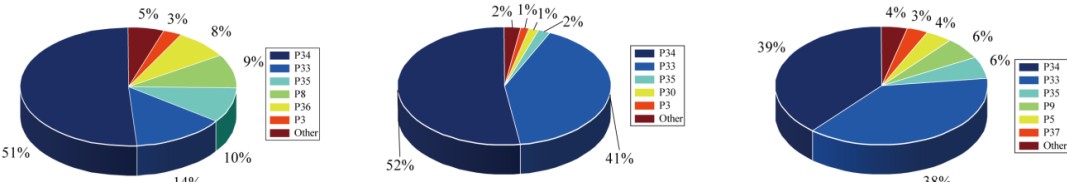

**Figure 2 The relative importance of the top 10 parameters to the target variables obtained by Sobol' sensitivity analysis.**
**The percentage of each parameter is the ratio between the total order Sobol' index of $\theta_i$ and the sum value of all the**
**parameters $\theta$. Each slice of the pie chart indicates the extent to which the changes in parameter $\theta_i$ can explain the total**
**variance of the model outputs.**
**3.2 Comparison of the Parameter Optimization Results**
The particle transitions, evolution, and optimized results of specific parameters in the three optimization scenarios
constrained by *LE* and *NEE* observations are shown in Table 1 and Fig. S7 to S9. Compared to the default values, the
optimized values of almost all the parameters have changed significantly (especially the two most sensitive parameters
P34 and P33), while very few parameters have remained unchanged (e.g., P36 and P30). At the same time, the posterior
distributions of the common parameters (i.e., P34, P33, and P35) in the three optimization scenarios are quite different
(Fig. 3). For example, the optimal solution of P34 toward two extremes in *Opt_LE* (195.79) and *Opt_NEE* (58.86) in
the prior range ([10,200]). Based on the total error minimization principle, the multi-objective simultaneous
optimization algorithm makes trade-offs in the simulation performance of the two target variables and calibrates
parameter P34 to an intermediate value (82.19). The optimization of parameter P33 is similar (*Opt_LE*: 0.0761,
*Opt_NEE*: 0.0702, *Opt_ALL*: 0.0735). However, this does not mean that the multi-objective optimization values of all
the parameters will be between the two single-objective optimization values. For example, the optimized value of P35
in *Opt_ALL* (8.87) is greater than that in *Opt_LE* (6.62) and *Opt_NEE* (8.52). This is because the simulation
performance of the model depends not only on the value of a single parameter but also on the combination effect
between parameters. Therefore, the calibrated values of the model parameters must be derived from the simultaneous
multi-objective optimization rather than from single-objective optimization or the comparison of the optimized values
of multiple single-objective optimizations.
**Table 1. The default and optimized values of the most sensitive parameters in the three optimization scenarios: *Opt_LE*,**
***Opt_NEE*, and *Opt_ALL*. The optimized results are the median values of the posterior distributions obtained by the PEM-**
**SMC algorithm.**

| | *Opt_LE* | | | *Opt_NEE* | | | *Opt_ALL* | |
|---|---|---|---|---|---|---|---|---|
| Para. | Default | Optimized | Para. | Default | Optimized | Para. | Default | Optimized |
| P34 | 100 | 195.79 | P34 | 100 | 58.86 | P34 | 100 | 82.19 |
| P33 | 0.08 | 0.0761 | P33 | 0.08 | 0.0702 | P33 | 0.08 | 0.0735 |
| P35 | 9 | 6.62 | P35 | 9 | 8.52 | P35 | 9 | 8.87 |
| P8 | 131.88 | 472.01 | P30 | 0.3 | 0.3 | P9 | 207.34 | 116.62 |
| P36 | 0.01 | 0.01 | P3 | 0.43 | 0.35 | P5 | 5.77 | 4.56 |
| P3 | 0.4348 | 0.35 | | | | P37 | 0.5 | 0.67 |



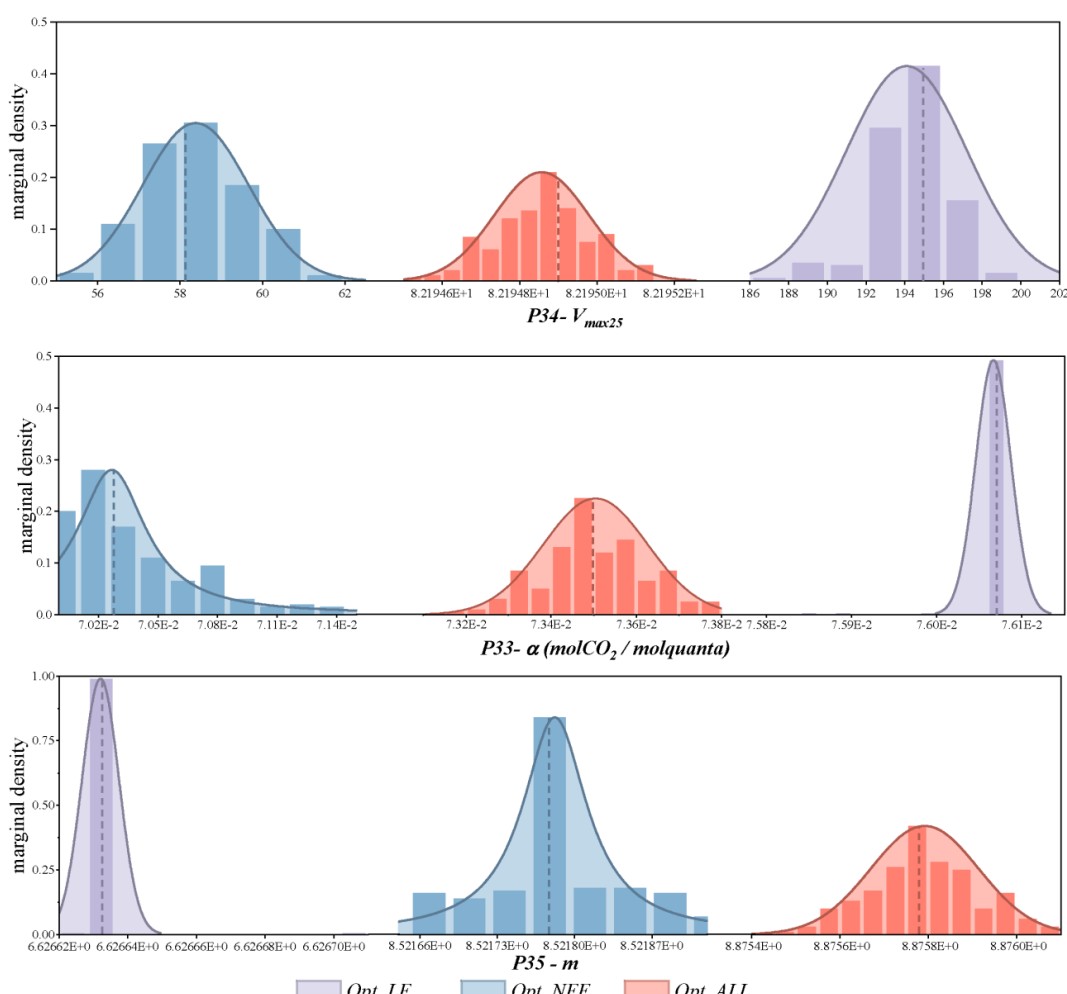

**Figure 3 The posterior distributions of the three sensitive parameters in the three optimization scenarios. The dashed line**
**in each graph denotes the median value of the posterior parameter distribution for each optimization scenario.**
**3.3 Comparison of the Optimization Effectiveness**
Three statistical metrics are used here to characterize the global effectiveness of the three optimized parameter
combinations in improving the performance of the CoLM in simulating *LE* and *NEE* (Fig. 4a–c). In addition, we
present the observations and simulations of *LE* and *NEE* under the different parameter combinations to show the
characterization of detailed changes in the two target variables in Fig. 4d–e.

By comparing *Control* and *Opt_LE*, it can be found that the performance of the CoLM in simulating LE can be
significantly improved by applying the optimized parameters of *Opt_LE*, as evidenced by the RMSE decreasing by
18.05 W/m2 and the NSE and R increasing by 0.18 and 0.07, respectively. The default parameter set for the CoLM
significantly underestimates LE, while the *Opt_LE* optimized parameter set significantly reduces the difference





between the simulated and observed *LE* (Fig. 4d). Meanwhile, we note that the simulation performance of the CoLM
for *NEE* was sacrificed by taking the optimized values of *Opt_LE*. From the statistical indicators, the R between the
simulated and observed *NEE* slightly increases by 0.09, but the RMSE increases by 6.94 μmol m-2 s-1 and the NSE
decreases to −1.09. From the values, the simulated NEE in the daytime (the negative value in Fig. 4e) under *Opt_LE*
is nearly three times lower than the observed values.

By comparing *Control* and *Opt_NEE*, it can be seen that the simulation performance for *NEE* does not improve
significantly after the single-objective optimization of *NEE*, as evidenced by the RMSE decreasing from 7.15 μmol m-
2 s-1 to 6.91 μmol m-2 s-1 and the NSE and R slightly increasing by 0.03 and 0.03, respectively. The simulated *NEE*
under the *Opt_NEE* parameter combination is essentially the same as that under *Control*, suggesting that the simulation
performance of the model for *NEE* cannot be improved by parameter calibration alone. Unlike *Opt_LE*, which
improves *LE* at the expense of the *NEE* simulation, *Opt_NEE*'s optimized parameter combination provides a slight
improvement in *LE* simulation performance (the RMSE is decreased by 1.22 and the NSE and R increase by 0.01).

Compared to single-objective optimization (*Opt_LE* and *Opt_NEE*), multi-objective optimization (*Opt_ALL*) can
simultaneously take into account the enhancement of the simulation performance for multiple variables. Compared to
*Control*, the *Opt_ALL* optimized parameters decrease the RMSE of *LE* and *NEE* by 7.2 W/m2 and 0.19 μmol m-2 s-
1, respectively, and increase the NSE of *LE* and *NEE* by 0.07 and 0.02, respectively (Fig. 4). Although the optimized
parameters of *Opt_ALL* are not as good as those of *Opt_LE* in improving the underestimated LE, it does not lose the
simulation accuracy for *NEE*. Comparing *Opt_NEE* with *Opt_ALL*, although they both improve the simulation
performance for LE and *NEE*, the latter's simulation accuracy for *LE* is significantly higher than that of the former.
In summary, multi-objective simultaneous optimization can improve the simulation performance for specific variables
without compromising the simulation accuracy of the other objective variables.



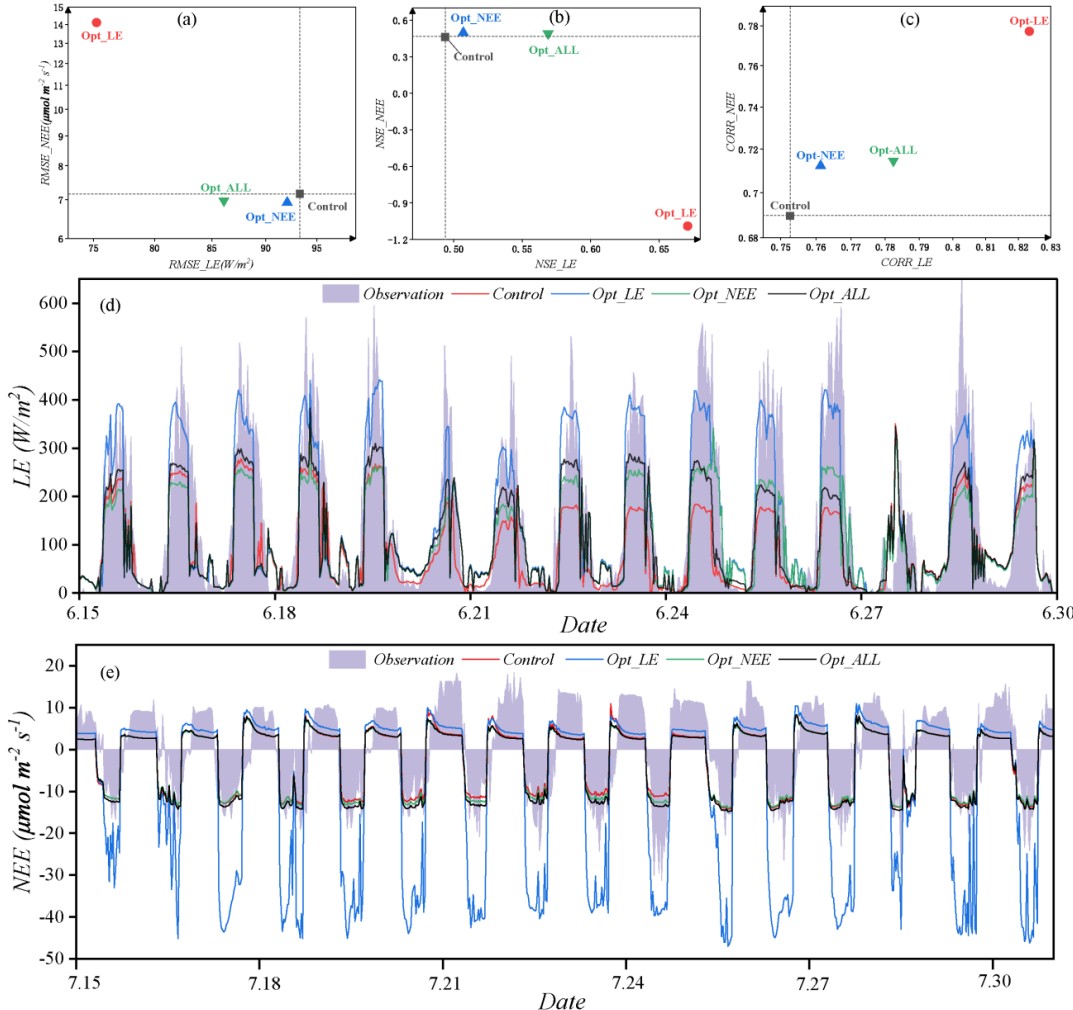

**Figure 4 Comparison of the performance of the CoLM in simulating LE and NEE under the four parameter schemes (*Control*, *Opt_LE*, *Opt_NEE*, and *Opt_ALL*) using three statistical metrics: (a) RMSE, (b) the Nash efficiency coefficient (NSE), and (c) the Pearson's correlation coefficient (R). Control denotes the default parameter values, while *Opt_LE*, *Opt_NEE*, and *Opt_ALL* represent the optimized values of the parameters under single-objective optimization of *LE*, single-objective optimization of *NEE*, and simultaneous multi-objective optimization of *LE* and *NEE*, respectively. The differences between the half-hourly observed and simulated (d) *LE* and (e) *NEE* in the four parameter schemes over a half-month period are also displayed.**

## 4. Discussion

### 4.1 The Advantages of the PEM-SMC Algorithm

This paper has introduced an enhanced PEM-SMC algorithm anchored in the Bayesian framework, which integrates

prior distributions with observational data to closely approximate the posterior distributions of the parameters. Facing

the challenge of the posterior distribution's analytical intractability, the proposed approach adopts an SMC method,





representing parameter distributions with a sequence of random particles that are iteratively updated through strategies
such as resampling to achieve a closer alignment with the actual posterior distribution. To mitigate the reduction in
particle diversity that can result from SMC resampling, a differential evolution mutation operator is introduced, aimed
at boosting the search efficiency within the parameter space. Consequently, this revised PEM-SMC algorithm not only
maintains the particle diversity but also optimizes the computational efficiency, surpassing its predecessor (Zhu et al.,
2018). Moreover, its capacity to yield a comprehensive posterior distribution, as opposed to the mere point estimations
offered by the deterministic parameter estimation techniques, significantly bolsters the robustness of the parameter
estimation (Thiemann et al., 2001; Jeremiah et al., 2011). This feature renders it particularly advantageous for total
uncertainty assessment, covering model parameters, inputs, and structural uncertainties.

Furthermore, the revised PEM-SMC algorithm effectively integrates the analytical power of Bayesian theory with the
flexibility of the SMC framework, demonstrating significant potential for practical applicability and structural
scalability. By applying Bayesian theory, it incorporates information from multiple sources, including prior knowledge
and observational data, into a unified analytical framework and expresses this information in a rigorous mathematical
form (Equations 14 and 16). Compared to the traditional metaheuristic multi-objective optimization algorithms (Deb
et al., 2002; Mirjalili et al., 2016; Xue et al., 2012), this framework solves multi-objective optimization problems more
directly through the joint probability distribution, avoiding the complexities of balancing multiple objectives via cost
functions. In addition, the PEM-SMC algorithm merges the strengths of the SMC and MCMC methods into a flexible
framework for structural extension (Speich et al., 2021), facilitating the design of adaptive transition kernels, effective
particle diversity enhancement strategies, and efficient intermediary proposal distributions.
**4.2 Differences Between Single- and Multi-Objective Optimization**
The distinction between the single- and multi-objective optimization strategies is first manifested in the parameter
estimation. This research has revealed a significant phenomenon: a single parameter can necessitate different optimal
values depending on the target variable. For instance, in the *Opt_LE* optimization scenario, the optimal value of
parameter P34 is nearly double its default setting, whereas in the *Opt_NEE* scenario, it is halved (Table 1). This
discrepancy arises from P34's (i.e., the maximum rate of carboxylation at 25 ℃, $V_{max25}$) dual role in the leaf stomatal
photosynthesis-conductance model, where it regulates both the rate of leaf photosynthesis and stomatal conductance
(see Sect. S3). To better align the simulation outcomes with actual observations, this parameter requires different
optimized values for the two target variables (i.e., *LE* and *NEE*). Specifically, as the simulated LE under the default
parameters falls short of the observed value, the optimization algorithm modifies P34 from 100 to 195.79, thereby
enhancing the leaf photosynthesis and reducing the stomatal resistance. Conversely, to minimize the discrepancy
between the simulated and observed *NEE*, the value of P34 is required to be halved, leading to a conflict in P34's
optimal values for both the *LE* and *NEE* simulations. Therefore, multi-objective optimization emerges as a key strategy
to balance the optimization performance of disparate target variables. In practice, the value of P34 is adjusted to 82.19,
situated between the *Opt_LE* and *Opt_NEE* values, rendering this balanced value more congruent with the default
empirical value of 100 and physically sound. Gong et al. (2015) also reported similar findings, where the "bsw"





parameter in the CoLM exhibited high optimal values under the constraints of observed sensible heat, latent heat, and
soil moisture, but showed low optimal values when constrained by upward long-wave radiation, net radiation, and soil
temperature observations. In fact, due to the complex interactions of the processes within LSMs, there are many such
"contradictory" parameters that can simultaneously affect multiple output variables. Although these sensitive
parameters can vary across different LSMs, ecological processes, or ecosystems, a common phenomenon is evident:
to maximize the reduction of discrepancies between the simulated values and observations across multiple target
variables, these parameters often demonstrate conflicting optimal values.

Furthermore, the difference between single- and multi-objective optimization is particularly evident in enhancing the
simulation performance across various output variables. While single-objective optimization, such as targeting solely
on *LE*, can improve the accuracy for that specific variable, it can adversely affect the simulation performance for other
variables, such as *NEE*. This highlights the complex interactions of the numerous processes (such as radiative transfer,
energy exchange, water transition, carbon cycling, etc.) in LSMs, where the impact of optimizing a single output on
the others is unpredictable. In contrast, multi-objective optimization has been proven to be more effective in improving
model performance across multiple outputs, as evidenced by the simultaneous improvement of *LE* and *NEE* in the
*Opt_ALL* scenario. Therefore, comprehensively integrating the various available observational data for multi-
objective optimization is preferable for parameter calibration in complex models. It is noteworthy that multi-objective
optimization may not achieve as high an accuracy for individual variables as single-objective optimization. For
instance, the discrepancy in the LE simulation of the *Opt_ALL* scenario (Fig. 4a: RMSE = 86 W/m2) compared to the
*Opt_LE* scenario (Fig. 4a: RMSE = 75 W/m2) indicates a trade-off. However, this trade-off is justified, as it results in
a more balanced and overall enhanced model performance at the expense of a slight sacrifice in simulation accuracy
improvement. If necessary, this limitation can be mitigated by adjusting the weights in the objective weighting method
to prioritize certain variables. In summary, the multi-objective optimization strategy is recommended for calibrating
complex models with multiple interrelated outputs, as it not only ensures that the optimized parameter values adhere
to objective physical constraints, but also balances the simulation performance among the multiple outputs.
**4.3 Defects in the Model Structure**
It is imperative to acknowledge that the potential for enhancing a model's simulation accuracy through parameter
optimization critically hinges on the robustness of the model structure and the quality of the driving data (Duan et al.,
2006). Our findings indicate that the imposition of constraints on the model parameters based on the *NEE* observations
does not yield a significant increase in the simulation performance for *NEE*, particularly concerning the
underestimated nocturnal respiration (Figure 4). This limitation is not attributable to the deficiency of the optimization
algorithm, but rather to the inadequate representation of the soil respiration processes within the model. In the CoLM,
soil respiration is quantified based on the exponential empirical equation (i.e., $R = R_{10}e^{E_o\left(\frac{1}{283.15-T_o}-\frac{1}{T-T_o}\right)}$, where R
and T are the soil respiration and soil temperature, respectively; $R_{10}$ denotes the basal respiration at 10 ℃; and $E_o$ is
the active energy; Lloyd et al., 1994). Theoretically, this approach is potentially more appropriate for estimating soil
respiration over annual or more extended temporal scales (Raich et al., 2002; Chen et al., 2013). A substantial body
of literature corroborates the existence of a phase lag (hysteresis) between the temporal dynamics of soil temperature
and soil respiration at hourly and seasonal timescales (Tang et al., 2005; Liu et al., 2006; Riveros-Iregui et al., 2007;
Ma et al., 2020). Such a lag leads to the empirical representation of soil respiration that diverges from the precise
modeling requirements of LSMs at hourly to daily intervals for carbon cycle simulation. This discrepancy highlights
the necessity for a more holistic consideration of the processes, encompassing soil heat and moisture dynamics,
microbial decomposition, and canopy photosynthesis (Hanson et al., 2000; Ryan et al., 2005; Davidson et al., 2006).
Therefore, the CoLM could be strengthened by integrating more detailed mechanistic process modules, exemplified
by the soil autotrophic and heterotrophic respiration modules featured in the Community Land Model (CLM4.5 and
5.0) (Lawrence et al., 2019), to substantially improve the accuracy and robustness of terrestrial carbon cycle simulation.
Accordingly, the simple calibration of model parameters falls short of addressing the inherent structural inadequacies
in the CoLM's representation of the soil respiration process. Therefore, parameter calibration (optimization) should
be viewed as a multifaceted tool—not only does it enhance the local-scale applicability of the model, but it also plays
a crucial role in uncovering the model's structural deficiencies and providing guidance for model refinement and
development.
**4.4 Limitations and Future Work**
This research has two principal limitations. Firstly, the PEM-SMC algorithm faces constraints in execution time, as
each evolutionary iteration of the particle swarm necessitates running the original dynamic model to evaluate the
likelihood function values of the different parameter combinations. Consequently, the cumulative runtime of the PEM-
SMC algorithm is quantified as $3 \times N_p \times S \times T$ (where $N_p$ represents the number of particles, $S$ is the number of
evolutionary iterations, and $T$ is the duration of a single dynamic model run), resulting in a computational demand
exceeding ten thousand operations. In response, our focus will be on refining the execution mode (parallel computing),
enhancing the sequential characteristics, and optimizing the resampling mechanisms, which are all aimed at
strengthening the efficiency of the PEM-SMC algorithm for complex model parameter optimization.

Secondly, the proposed approach for multi-objective optimization essentially converts the different objective variables
into a single-objective framework through a goal-weighting strategy. However, despite equalizing the weights for two
variables (*LE* and *NEE*), the subjectivity and uncertainty in the weight allocation could potentially restrict the diversity
of the optimal solutions. Therefore, in the future, we will work on developing a Bayesian-inspired multi-objective
parameter estimation algorithm. This algorithm will synergize the autonomous optimization capabilities of the PEM-
SMC algorithm with the non-dominant and diverse characteristics inherent in Pareto optimal solution theory, thereby
substantially augmenting the effectiveness and applicability of the PEM-SMC algorithm in complex multi-objective
optimization scenarios.





**5 Conclusions**

In this study, we employed the revised PEM-SMC algorithm for single- and multi-objective optimization of the sensitive parameters in the water-carbon process of the CoLM and conducted a comparative analysis of different optimization strategies concerning parameter estimation, model performance enhancement, and applicability reliability. The key findings include:

Firstly, the revised PEM-SMC algorithm demonstrates a robust ability to tackle the multi-dimensional, multi-objective parameter optimization challenge for complex dynamics models. Secondly, significant differences were observed between single- and multi-objective optimization in parameter estimation. The optimization values of the three sensitive parameters present conflicts after single-objective optimization, while the values after multi-objective optimization appear more rational. Moreover, the multi-objective optimization demonstrated superiority over single-objective optimization in enhancing the simulation performance for the multiple output variables. Although single-objective optimization can improve the simulation performance for specific objectives, it can adversely affect the other target variables. For instance, optimizing for LE reduced the RMSE of the simulated and observed *LE* by 20%, but increased the RMSE of *NEE* by 97%. Conversely, the multi-objective optimization concurrently improved the simulation performance for both LE and NEE, evidenced by decreases in RMSE for *LE* and *NEE* of 7.2 W/m2 and 0.19 μmol m-2 s-1, respectively. Finally, through the parameter optimization process, we identified the structural deficiencies in the CoLM's soil respiration calculations. Consequently, we suggest that the CoLM modeling community consider integrating more precise mechanistic process models to enhance the accuracy and robustness of terrestrial carbon cycle simulation.

**Code and Data Availability**

The detailed information for RU-FY2 can be accessed via FLUXNET at https://fluxnet.org/sites/siteinfo/RU-Fy2. The flux data employed in this research were obtained from FLUXNET2015, available at https://fluxnet.org/. The sensitivity analysis software package, PSUADE version 1.7.8a, is accessible at https://computing.llnl.gov/projects/psuade/software. The CoLM codes (2014 version) can be found through the Land-Atmosphere Interaction Research Group at Sun Yat-sen University, detailed at http://globalchange.bnu.edu.cn/research/models. Comprehensive access to the data, code (inclusive of the CoLM model program, sensitivity analysis software package, and the revised PEM-SMC program), and associated outcomes from this study are available at https://doi.org/10.5281/zenodo.10900461.

**Author contributions**

Under the guidance of GFZ, CX was responsible for conceptualization, algorithm development, data handling, and drafting the initial manuscript. YZ and KZ revised the initial draft of the manuscript and provided improvement suggestions. All authors discussed the results throughout the research period and approved the final version of the manuscript for publication.



**Competing interests**

The contact author has declared that none of the authors has any competing interests.

**Acknowledgments**

This work received funding from the National Natural Science Foundation of China (grant nos. 42171019) and the Key Science and Technology Foundation of Gansu Province (grant no. 23JRRA1025). We would like to express our sincere gratitude to the principal investigators and their teams for providing the site observation datasets used in this study. In addition, we acknowledge the support from the Supercomputing Center of Lanzhou University. We declare that there are no financial or personal relationships with other individuals or organizations that could inappropriately influence our work.

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
