# Peer review of "Comparing the Impacts of Single- and Multi-Objective"

_Hydrology and Earth System Sciences, 2024_

## Referee Comment (RC1)

**Review of the manuscript « Comparing the impacts of single and multi-objective optimization on the parameter estimation and the performance of a land surface model » by Xu et al.**

General comment:

The paper proposed by Xu et al investigates several issues related to sensitivity analysis (SA) and optimization using the Land Surface Model CLM. The final objective is to study the effect of using a single or several variables during the optimization process on parameter estimation and the overall performance of CLM applied on an ICOS site located in Russia. Before performing optimization, sensitivity analysis is performed using 4 approaches to identify the parameters that mostly impact the simulated variables. Optimization is then performed in a single or mutli-objective mode using the PEM-SMC algorithm that was specifically adapted to reduce the computational burden and make such an optimization possible.

I believe that the paper proposed is interesting. Considering the information provided in the introduction on the increasing trend to use multi-objective optimization, I feel like the outcomes of the paper are not very significant. This paper still provides a good illustration of why multi-objective optimization should be preferred to improve the robustness and the prediction capabilities of LSM. Overall, I found that the methodological aspects and explanations on the choices made could have been more developed. Although CLM is a LSM widely used in the community, basic aspects on the equations and parameters behind should be presented somewhere. Without this information, it is hardly possible to clearly understand the sensitivity analysis results and the parameter optimization. The paper also lacks schemes that would greatly help in understanding the methods used. I think this paper should be improved significantly before being considered for publication in HESS. I would like the following comments to be considered or answered if possible:

Major comments:

- As mentioned above, the paper is not self-consistent as no information on CLM - equations and parametrization – are provided. In my opinion, the paper should be reshaped to include a part dedicated to the presentation of CLM. Furthermore, the name used in the paper – CoLM – should be changed throughout the paper and turned into CLM to avoid confusion.

- I don't get why 3 qualitative sensitivity analysis approaches are used prior to the Sobol's analysis. From Figure 1, it seems that MOAT alone could be sufficient to identify the most sensitive parameters to be used in the following.  In my opinion, the need of multiple qualitative approaches, their potential complementarity and what kind of different information they can bring in should be detailed and explained more clearly. The use of 3 methods rather than one makes it more difficult for the reader to understand the overall method.

- If the use of the 3 approaches is relevant, the description of each approach should be improved to better explain its own interest for sensitivity analysis. The sizes of the different samples seem to be set arbitrarily. Maybe justifications – that are not only related to the computation burden – should be given as it can impact the performance of the sensitivity analysis.

- The description of the overall approach – presented from L189 to L212 – should be improved. As it stands in this version, sensitivity analysis and optimization are mixed together which is rather hard to catch. I think a scheme is highly needed here. And I also think that the authors

should more clearly stands that the target variables are NRMSEs computed with LEE/NEE/both.

- There are also some discrepancies between what is presented L189 to 212 and what is presented afterwards. It is stated L205 that 10 parameters are selected for SA when less parameter are kept in the application example. It is said that the optimization is guided by Sobol's analysis. Does that mean that some parameters are removed after Sobol's indices are computed?

- The technical aspects of part 2.3 are very hard to follow. Once again, I fell like a scheme could help understanding what is proposed and done.

- It's not clear how many particles/set of parameters are kept during the optimization process. I think this should be clearly specified somewhere. The way the values for non-sensitive parameters are set should also be clearly explained.

- After the SA results are presented, I think the physical meaning of the sensitive parameters should be explained. In my opinion, SA brings insights on how a model works. This aspect is rather poorly developed in the paper. This could greatly help for the analysis of the results, especially to understand the different values obtained after single/multiple optimization.

- After the optimization, some optimized parameters – P36 and P3 - reach one of the bounds of its variation interval. In my opinion, this is a bit troublesome and this question the way the bounds of the intervals were chosen.

Specific comment:

- In the abstract and conclusion, the impact of efficiency is sometimes in % and sometimes in raw values. I think it's more convenient and easier to use % everywhere.
- L52: what's the difference between LSM and soil-vegetation-atmosphere coupled models?
- L135: please specify the signification of delta here?
- L187: extensive dataset (104 or 105 or more): I guess it 10^4 and 10^5?
- Fig 3: change the values on the x-coordinates. Not easy to read.

---

## Author Comment (AC1)

**General Comment:**

The paper proposed by Xu et al investigates several issues related to sensitivity analysis (SA) and optimization using the Land Surface Model (CLM). The final objective is to study the effect of using a single or several variables during the optimization process on parameter estimation and the overall performance of CLM applied on an ICOS site located in Russia. Before performing optimization, sensitivity analysis is performed using 4 approaches to identify the parameters that mostly impact the simulated variables. Optimization is then performed in a single or multi-objective mode using the PEM-SMC algorithm that was specifically adapted to reduce the computation burden and make such an optimization possible.

**Response:**

We appreciate the reviewer's positive and constructive feedback, which will assist us in further improving our work. Below, we outline our planned responses to the issues raised by the reviewer and the specific changes we intend to implement in the revision.

**Major Comments:**

**Comment #1:** As mentioned above, the paper is not self-consistent as no information on CLM- equations and parameterization-are provided. In my opinion, the paper should be reshaped to include a part dedicated to the presentation of CLM. Furthermore, the name used in the paper-CoLM-should be changed throughout the paper and turned into CLM to avoid confusion.

**Response:**

Thank you for your valuable suggestions. We recognize the importance of including equations and parameterization details for latent heat flux (LE) and net ecosystem exchange (NEE) in the Common Land Model to clarify the physical basis of the selected sensitive parameters and the mechanisms underlying their optimized values. Accordingly, We will add descriptions of the LE and NEE equations in Section 2.1 of the revised manuscript. Regarding the abbreviation, while "CLM" was initially used by Dai et al. (2003), subsequent studies have adopted "CoLM" to avoid confusion with the Community Land Model, which is also widely referred to as "CLM." Therefore, we have retained the abbreviation "CoLM" in our manuscript. However, if you believe that

"CLM" should be used, we are open to making this change in the next revision.

**Reference:**

Dai, Y., Zeng, X., Dickinson, R. E., Baker, I., Bonan, G. B., Bosilovich, M. G., ... & Yang, Z. L. (2003). The common land model. Bulletin of the American Meteorological Society, 84(8), 1013-1024.

**Comment #2:** I don't get why 3 qualitative sensitivity analysis approaches are used prior to the Sobol's analysis. From Figure 1, it seems that MOAT alone could be sufficient to identify the most sensitive parameters to be used in the following. In my opinion, the need of multiple qualitative approaches, their potential complementarity and what kind of different information they can bring in should be detailed and explained more clearly. The use of 3 methods rather than one makes it more difficult for the reader to understand the overall method. If the use of the 3 approaches is relevant, the description of each approach should be improved to better explain its own interest for sensitivity analysis.

**Response:**

Qualitative sensitivity analysis methods typically evaluate the importance of input parameters on model output by comparing outputs under different combinations of inputs with relatively few samples. However, complex models often involve characteristics like nonlinearity and parameter interdependence, and different methods may emphasize various aspects of these features. Relying on a single method could overlook or misjudge certain parameters. For instance, in the Opt_LE scenario (Figure 1), MOAT did not identify sensitive parameter P36, while MARS did; both MOAT and MARS identified P8, but DT did not. Combining multiple methods can enhance the robustness of the analysis, reduce errors, and yield more reliable results. Given the previous limitations in our description, we will provide a more detailed explanation of the strengths and differences of these three methods, emphasizing their complementarity, in Section 2.2 of the revised manuscript:

(1) Delta test (DT) method: While it estimates complex nonlinear relationships, it may not identify the most sensitive parameters. Combining DT with MARS and MOAT helps overcome this limitation.

(2) Multivariate adaptive regression splines (MARS) method: Effective in handling nonlinear relationships and interactions, but it can sometimes create overly complex models. Integrating MARS with DT and MOAT helps to clarify the impact of each parameter.

(3) Morris method: Provides a global perspective, offering a comprehensive evaluation of all parameters and reducing biases inherent in local sensitivity analyses.

**Comment #3:** The size of the different samples seem to be set arbitrarily. Maybe justifications – that are not only related to the computation burden – should be given as it can impact the performance of the sensitivity analysis.

**Response:**

In this study, the sample size for all qualitative analysis methods was determined based on the findings of Li et al. (2013), who conducted a sensitivity analysis on 40 parameters of the CoLM model. For the DT and MARS methods, sample sizes of 200, 400, and 1000 (i.e., 5, 10, and 25 times the number of parameters, respectively) were evaluated. For the MOAT method, samples were typically set as multiples of n+1, where n is the number of parameters; hence, sample sizes of 205, 410, and 1025 were examined. Their results indicated that a sample size of 400 (10 times the number of parameters) was sufficient for screening the 40 parameters of the CoLM model. Therefore, this study utilizes 400 samples for DT and MARS methods and 410 for MOAT. In the revised manuscript, we will include additional tests and comparisons using different sample sizes (5, 10, and 25 times the number of parameters) to ensure the robustness of the sensitivity analysis results. For the quantitative sensitivity analysis (Sobol'), the sample size was set to 100,000, consistent with the typical range reported in previous studies ($10^4$ to $10^5$). For instance, Rosolem et al. (2012) used 45,000 model runs to evaluate the Sobol' sensitivity indices of 42 parameters in the Simple Biosphere 3 (SiB3) model, while Zhang et al. (2013) employed 60,000 model runs to study the sensitivities of 28 parameters in the Soil and Water Assessment Tool (SWAT) model using the Sobol' method.

**Reference:**
Li, J., Duan, Q. Y., Gong, W., Ye, A., Dai, Y., Miao, C., ... & Sun, Y. (2013). Assessing

parameter importance of the Common Land Model based on qualitative and quantitative sensitivity analysis. *Hydrology and Earth System Sciences*, *17*(8), 3279-3293.

Rosolem, R., Gupta, H. V., Shuttleworth, W. J., Zeng, X., & de Gonçalves, L. G. G. (2012). A fully multiple-criteria implementation of the Sobol′ method for parameter sensitivity analysis. *Journal of Geophysical Research: Atmospheres*, *117*(D7).

Zhang, C., Chu, J., & Fu, G. (2013). Sobol″'s sensitivity analysis for a distributed hydrological model of Yichun River Basin, China. *Journal of Hydrology*, *480*, 58-68.

**Comment #4**: The description of the overall approach    presented from L189 to L212  should be improved. As it stands in this version, sensitivity analysis and optimization are mixed together which is rather hard to catch. I think a scheme is highly needed here. And I also think that the authors should more clearly stands that the target variables are NRMSEs computed with LEE/NEE/both.

**Response:**

In response to your suggestion, we will include a technical flowchart in the revised manuscript to more clearly illustrate the methods, processes, and metrics involved in the parameter sensitivity analysis. Additionally, we have revised the cost function expressions for both single-objective and multi-objective sensitivity analysis as follows:

*Given the varying magnitudes of the target variables (LE/NEE), we employed the normalized root-mean-square error (NRMSE) as the cost function, defined as:*

$$NRMSE_i = \frac{\sqrt{\Sigma_{t=1}^{T}\,S_i(t)-O_i(t))^2}}{\Sigma_{t=1}^{T}\,O_i(t)} \qquad (11)$$

*where $i$ represents the target variable (eg., LE or NEE), $T$ is the total number of simulations; and $S_i(t)$ and $O_i(t)$ represent the simulated and observed values of the target variables, respectively. For single-objective sensitivity analysis, the cost function is expressed as:*

$$F_i = NRMSE_i \qquad (12)$$

*where $F_i$ denotes the error evaluation of the target variable $i$ (e.g., LE or NEE). For multi-objective (LE+NEE) sensitivity analysis, the combined objective function can be expressed using a weighted sum of the individual objective functions. The simplest form is:*

$$F_{LE+NEE} = w_{LE} \cdot F_{LE} + w_{NEE} \cdot F_{NEE} \qquad (13)$$

*where $w_{LE}$ and $w_{NEE}$ are weights, typically set to 1 to indicate equal weighting.*

**Comment #5:** There are also some discrepancies between what is presented L189 to 212 and what is presented afterwards. It is stated L205 that 10 parameters are selected for SA when less parameter are kept in the application example. It is said that the optimization is guided by Sobol's analysis. Does that mean that some parameters are removed after Sobol's indices are computed?

**Response:**

We apologize for any confusion. Before addressing your question, I will briefly explain the differences between qualitative and quantitative sensitivity analysis. Qualitative analysis is typically used for initial parameter screening, providing a rough ranking of parameter influence. In contrast, quantitative analysis offers precise quantification of each parameter's contribution to the model output and its interactions with other parameters, though it requires higher computational costs, especially in high-dimensional parameter spaces. Direct application of quantitative sensitivity analysis (e.g., the Sobol' method) to complex models can lead to inefficient use of computational resources and unnecessary complexity. In practice, the number of parameters analyzed using Sobol' methods is usually limited to 10.

Our sensitivity analysis is therefore conducted in two stages. First, qualitative analysis (using DT/MARS/MOAT methods) is performed with a smaller sample size to identify the 10 most sensitive parameters from an initial set of 40 (see Fig.1), thereby making the subsequent quantitative analysis more targeted and effective. In the second stage, Sobol' quantitative analysis is applied to these 10 parameters. Parameters for optimization are selected based on the criterion that their cumulative relative importance exceeds 95%, indicating that these parameters account for 95% of the explained variance (see Fig.2). As a result, some parameters identified in the qualitative analysis are excluded after the Sobol' analysis.

**Comment #6:** The technique aspects of part 2.3 are very hard to follow. Once again, I feel like a scheme could help understanding what is proposed and done.

**Response:**

Thank you for your suggestion. Previously, I had attempted to visually represent the

process of the PEM-SMC algorithm, as shown in the figure below. In the revised manuscript, I will include a schematic diagram of the improved version of this algorithm to enhance clarity.

[Figure]

Figure R1 The schematic diagram of the PEM-SMC algorithm process.

**Comment #7:** It's not clear how many particles/set of parameters are kept during the optimization process. I think this should be clearly specified somewhere. The way the values for non-sensitive parameters are set should also be clearly explained.

**Response:**

In the PEM-SMC optimization algorithm, the initial number of particles (e.g., N=200) remains fixed throughout the iterative process, while the parameter values represented by each particle are continually updated. In Section 2.3 of the revised manuscript, we will provide a more detailed explanation of the role of particles in PEM-SMC: each particle corresponds to a specific point within a multidimensional parameter space, and through the evolution of these particles and their associated weights, the algorithm progressively converges towards the true posterior distribution of the parameters. For the optimization of each target variable, multiple sensitive parameters are considered, each with a specific range of possible values. Each particle represents a unique combination of these parameter values.

Furthermore, the non-sensitive parameters were assigned based on the model's default settings, which take into account factors such as vegetation type, soil characteristics, or

data derived from empirical studies and literature. In the revised manuscript, we will include Table 1 in the Supporting Information, where the default values for the 40 predefined parameters specific to this site will be provided. Additionally, in Section 2.3 of the revised manuscript, we will elaborate on how the non-sensitive parameters were determined using the model's default settings.

**Comment #8:** After the SA results are presented, I think the physical meaning of the sensitive parameters should be explained. In my opinion, SA brings insights on how a model works. This aspect is rather poorly developed in the paper. This could greatly help for the analysis of the results, especially to understand the different values obtained after single/multiple optimization.

**Response:**

Thank you for your valuable suggestion. We acknowledge that the sensitivity analysis results are closely tied to the model's representation of relevant processes and its parameterization scheme. For instance, the two most sensitive parameters for calculating LE and NEE, P33 and P34, correspond to the quantum efficiency and the maximum carboxylation rate of vegetation leaves at 25 ℃ in the leaf stomatal photosynthesis-conductance module. These parameters directly affect the calculation of net photosynthetic rate and stomatal conductance. In the original manuscript, our explanation of their physical significance was lacking. In response, we will provide a more detailed exploration of the physical basis of these sensitive parameters, incorporating the CoLM model's parameterization scheme in Section 3.1 of the revised manuscript to improve both the depth and clarity of the analysis.

**Comment #9:** After the optimization , some optimized parameters – P36 and P3 – reach one of the bounds of its variation interval. In my opinion, this is a bit troublesome and this question the way the bounds of the intervals were chosen.

**Response:**

The optimized values of P36 and P3 have indeed reached the boundaries of their respective ranges. However, this does not necessarily indicate a flaw in the optimization process or its outcomes. First, the parameter ranges were reasonably established based on experimental data, literature, and field conditions, ensuring compliance with

physical constraints rather than being arbitrarily set. Second, the fact that these parameters reached their boundary values may suggest that these values represent the optimal solution, indicating the optimization algorithm identified the best configuration within the given range. Nevertheless, we also recognize that this outcome could be influenced by insufficient sample data or data uncertainty, potentially causing the mode to rely on boundary values for optimal fitting. In response, we will appropriately adjust the parameter ranges - either narrowing or expanding them – in the revised manuscript to explore the impact of these boundaries on the optimization results and to assess the robustness of the outcomes.

**Specific comment:**

**Comment #1:** In the abstract and conclusion, the impact of efficiency is sometimes in % and sometimes in raw values. I think it's more convenient and easier to use % everywhere.

**Response:**

Thank you for your suggestion. We have updated both the abstract and conclusion to consistently use percentages throughout.

**Comment #2:** L52: what's the difference between LSM and soil-vegetation-atmosphere coupled models?

**Response:**

Thank you for your question. Land Surface Models (LSMs) focus on simulating energy, water, and carbon exchanges between the land surface (including soil, vegetation, and snow) and the atmosphere. They are typically part of larger climate or weather models. In contrast, soil-vegetation-atmosphere coupled models integrate the interactions between soil, vegetation, and the atmosphere, capturing more complex feedback such as the effects of soil moisture and vegetation changes on atmospheric processes. While LSMs are often part of these coupled models, the latter provides a more comprehensive view of these interactions. If you believe it is better to combine the two model types in the manuscript, we can revise accordingly.

**Comment #3:** L135: please specify the signification of delta here?

**Response:**

Thanks for your question. The delta $\delta(\varepsilon)$ denotes the noise variance, which serves to estimate the error in the output $Y$ caused by random noise $\varepsilon$. The Delta Test (DT) method aims to quantify the noise by analyzing the differences between the nearest neighbors in the input space. In this context, $\delta$ measures the noise level present in the output. In essence, the DT method calculates the noise variance $\delta(\varepsilon)$ by comparing the output values of nearest neighbors with those of the corresponding original data points. By minimizing $\delta(\varepsilon)$, the method identifies the subset of input parameters that most significantly influences the output.

**Comment #4:** L187:extensive dataset(104 or 105 or more):I guess it 10^4 and 10^5?

**Response:**

Thank you for pointing out this mistake. You are correct, it should be 10^4 and 10^5. I have made the necessary corrections in the revised manuscript.

**Comment #5:** Fig 3: change the values on the x-coordinates. Not easy to read.

**Response:**

Thank you for your valuable suggestion. In the revised manuscript, we will adjust Figure 3 by plotting the three distributions on a single, linearly-scaled x-axis. This modification will allow for a clearer comparison of the parameter distributions across the different optimization methods.

---

## Author Comment (AC2)

Thank you very much for doing this work. I think the paper could make a contribution to HESS after some work. I leave here my main comments, I hope they help to improve the manuscript.

**Response:**

Thank you for your valuable feedback. We appreciate your comments and will address each suggestion with corresponding revisions to improve the manuscript.

**Comment #1:** Literature review. I miss references to work of Jasper Vrugt, mainly to Vrugt et al., 2012 (the DREAM paper), you might have a look at the work of Carlo Alber about ABC(during 2015 and maybe reply rom Vrugt later on), Kavetski et al. 2018 and Fenizia et al 2018 look through the introduction+ references of these 2 papers;and the work from Prieto et al. 2021;2022 about hydrological mechanisms identification and the diagnostic metrics there in. Othersize the introduction is a bit repetitive and convoluted but it misses inormation, e.g. about the choose or development of different likelihoods (later on you assume a normal gaussian but I do not see the justification), model diagnostics metrics and why.

**Response:**

We appreciate your suggestion to reference Jasper Vrugt et al's 2012 work on the DREAM algorithm, as well as Carlo Albert's research on the ABC method, particularly his 2015 study and subsequent discussions. We also acknowledge the contributions of Kavetski et al. and Fenizia et al. in 2018. These studies are highly relevant to our topic, and we will review and incorporate them into the revised manuscript to strengthen the comprehensiveness of our literature review.

Regarding the likelihood function, your feedback is valuable. We currently use a Gaussian distribution but have not sufficiently explained its theoretical basis. In the revised manuscript, we will clarify the rationale for this choice, and its relevance to our algorithm, and provide a comparison with other commonly used likelihood functions to explain our final selection.

2. benchmark: I am missing a benchmark to compare. Maybe, one good idea might be to use the package from Vrugt for DREAM as benchmark – I am aware the author had everything ready to be applied.

**Response:**

Regarding the benchmark testing of the PEM-SMC algorithm, we evaluated its effectiveness through two experiments, as presented in Supplementary Information S1, "Evaluating the effectiveness and efficiency of the revised PEM-SMC algorithm". However, we recognize that these internal tests primarily demonstrate the algorithm's capability to estimate the target posterior distribution. A more comprehensive evaluation of its performance, including residual characteristics, confidence intervals, and computational speed, requires comparison with other algorithms, which is currently missing. Therefore, in the revised manuscript, we will conduct a comparative analysis between the PEM-SMC and DREAM algorithms to provide a more thorough evaluation of PEM-SMC's performance. Additionally, we will also reassess the structure and framework of the PEM-SMC algorithms in relation to the DREAM algorithm.

3. For the equations I suggest to use a properly maths notation. At least for me is helpful and I am sure for readers too. E.g. why is everything italic? I suggest you to differenciate vectors, matrices, random variables, etc (ie bold, capital letters, and so on). In Prieto et al., 2019; 2021; 2022 you can find examples for this and in whatever paper from Vrugt I am sure too.

**Response:**

Thank you for your suggestion. We acknowledge the current manuscript's limitations in the representation of equation symbols and variables, particularly the use of italics for all elements. In response, we will revise the manuscript to adopt a more standardized mathematical notation. Specifically, we will use bold italics to distinguish vectors and matrices, and uppercase letters for random variables, to enhance clarity and readability. We will refer to the works of Prieto et al. (2019, 2021, 2022) and Vrugt's research to ensure consistent and standardized symbol usage for the benefit of the readers.

4. Posterior pdf: please for the posterior of the parametere use the full Bayes equation and then say that the left hand is proportional to the right hand so that non Bayesian can follow it. Indeed this is meet bacause you use only one model (eg see prieto et al., 2021, 2022).

**Response:**

Thank you for your suggestion. Following the work of Prieto et al. (2021), we have revised the posterior probability density function (PDF) of the parameters in the original manuscript and provided a more detailed explanation of the full Bayesian equation. To improve clarity, we have also incorporated a conceptual model representation, further elaborating on how the likelihood function, when combined with observed data and prior knowledge, results in the derivation of the posterior parameter distribution. The revised section is as follows:

*In this study, we address the inference of model parameters in CoLM using observed data $\tilde{q} = (\tilde{q}_t; t = 1, \dots, N_t)$, which represents a time series of target variable observations (e.g., LE and NEE) of length $N_t$. Within the Bayesian framework, model parameters are conceptualized as probabilistic variables, and the posterior distribution of the parameters, $p(\theta \mid \tilde{q}, G, \tilde{x}, s_0)$, is expressed as:*

$$p(\theta \mid \tilde{q}) = \frac{p(\theta)p(\tilde{q}|\theta)}{p(\tilde{q})} = \frac{p(\theta|G,\tilde{x},s_0)p(\tilde{q}|\theta,G,\tilde{x},s_0)}{p(\tilde{q}|G,\tilde{x},s_0)}$$

*Where $p(\theta)$ is the prior distribution of its parameters over its feasible domain, $p(\tilde{q}|\theta)$ is the likelihood function associated with the probability mode and $p(\tilde{q})$ is referred to as Bayesian Model Evidence (BME) or Marginal Likelihood. The BME term is generally the normalization constant and is not required for parameter inference. The model structure $G$, forcing data $\tilde{x} = (\tilde{x}_t; t = 1, \dots, N_t)$, and initial conditions $s_0$ are treated as fixed in this study.*

5. – concern: why the likelihood is a normal likelihood? could you please justify and then analize the residuals of the posteriors? are you also meaning that the vairables are independent and then the likelihoods can bu multiplied? I suggest you to have a look at ABC hear just in case it can help.

**Response:**

Thank you for your valuable feedback and suggestions.

After conducting a posterior analysis, we found that the residuals of the LE and NEE variables align more closely with a t-distribution, rather than the initially assumed normal distribution (Figure 1). The normal distribution was our initial choice due to its

computational simplicity and widespread use. However, the t-distribution, with its heavier tails, is more sensitive to outliers and may better fit our current dataset. While there are notable differences between the likelihoods of the normal and t-distributions, particularly in handling extreme values, their roles in parameter optimization are quite similar. Therefore, we believe that this discrepancy is unlikely to significantly impact the optimization results. Nevertheless, to ensure the robustness of our findings, we plan to use the t-distribution likelihood function in future work and compare its performance with that of the normal distribution to validate this assumption.

We assume that the observations are independent, allowing the likelihood function to be expressed as a product of the probabilities for each independent observation. This independence assumption simplifies the computational complexity and ensures the interpretability of the model. Given the current structure and data of the model, this assumption appears reasonable. However, if dependencies between observations emerge in future models, we will revisit and adjust this assumption accordingly.

We also appreciate your suggestion regarding the use of Approximate Bayesian Computation (ABC). ABC is particularly useful in cases where the likelihood function is intractable, providing an effective approach for parameter estimation in complex models. Although we can compute the likelihood function directly in the current study, we will thoroughly review relevant literature and explore the potential application of ABC in more complex scenarios in future research.

[Figure]

Figure. R1 Comparison of the fitting performance between the normal distribution and t-distribution for the residuals of the LE and NEE target variables.

6. – diagnostic metrics: also, could you please take the advantage of doing probabilistic

analysis to evaluate the posterior pdf using probabilitic metrics to look at reliability, precision and bias – ie not only deterministic (related) metrics, this only inspects one side of the history. Based on this, the advantages highlighted on the discussion section could be more defended.

**Response:**

Thank you for your valuable feedback.

We recognize that assessing the model's performance deterministically using the optimal solution from the posterior distribution (e.g., the posterior median) diverges from the main objective of uncertainty analysis in the Bayesian framework. To address this, the revised manuscript will include the full Bayesian predictive distribution based on the entire posterior distribution of the parameters, along with associated confidence intervals. Additionally, we will incorporate probabilistic metrics and uncertainty evaluation methods, such as those grounded in scoring rules, to provide a more comprehensive assessment of the model's fit. This approach will not only capture historical data but also fully utilize the advantages of Bayesian probabilistic analysis, thereby improving the model's interpretability and enhancing the reliability of the results.

7. – for me the text is a bit confusing when talking about multiple objectives, I guess most of the readers tend to think about multiple objective functions which is not the case bacause there is 1 likelihood – other thing is that there are 2 target variables.

**Response:**

Thank you for your feedback. The use of the term "multiple objectives" in our discussion may have led to the misunderstanding that we were referring to multiple objective functions. However, in this context, "multiple objectives" refers to the two target variables in the model (LE and NEE), not to multiple optimization functions. We employed a single likelihood function to fit these two target variables simultaneously.

In a deterministic parameter optimization framework, multiple objective functions are typically employed to find an optimal solution that minimizes the residuals of each target variable. However, in the Bayesian framework, we can represent the likelihood for multiple target variables as the product of their respective likelihood functions,

achieving joint optimization. This method allows us to account for uncertainty while simultaneously addressing multiple target variables, with each variable's contribution represented by its respective likelihood function. This approach effectively integrates information from different target variables, ensuring that the model captures not only historical data but also the inherent uncertainty in the parameters.

To prevent further confusion, we will clarify the distinction between "target variables" and "objective functions" in the revised manuscript, and provide a more detailed explanation of how a single (or joint) likelihood function is used within the Bayesian framework to model multiple target variables.

8. – maybe a naïve question, but do you need all the SA methods in the main manuscript?

**Response:**

Thanks for your question. In this study, we employed a two-stage approach that combines three qualitative sensitivity analysis methods (DT, MARS, and MOAT) with one quantitative method (Sobol') to identify the most influential parameters in the CoLM model for LE and NEE simulations. This approach was necessary to balance computational efficiency with robust parameter identification.

First, qualitative methods use relatively small sample sizes (hundreds to thousands) to provide an initial ranking of parameter influence by comparing model outputs across different input combinations. Given the complexity of the model, which involves nonlinearity and parameter interactions, relying on a single method risks overlooking key parameters. By combining multiple qualitative methods, we enhance the robustness of the initial parameter screening.

Second, while quantitative methods like Sobol' provide precise estimates of parameter contributions and interactions, they are computationally intensive, especially for high-dimensional parameter spaces. To mitigate this, we first applied qualitative methods to reduce the parameter set from 40 to the 10 most sensitive parameters (see Figure 1), thereby reducing the computational burden. In the second stage, Sobol' analysis was used to further evaluate these 10 parameters, identifying those that explained over 95% of the variance for further optimization (see Figure 2).

In summary, this two-stage process efficiently integrates qualitative and quantitative

methods, allowing for robust identification of key parameters while minimizing computational complexity.

---

## Author Comment (AC3)

**Summary:** In this study, the authors applied a revised particle evolution Metropolis sequential Monte Carlo called PEM-SMC to single- and multi-objective optimization of the Common Land Model (CoLM) using measurements of the latent heat flux (LE) and net ecosystem exchange (NEE) from a typical evergreen needle-leaf forest observation site. The authors conclude that the revised PEM-SMC algorithm is a robust method for LE and NEE exhibit a trade-off, necessitating the estimation of these paraemters against LE and NEE data simultaneously.

**Evaluation:** This paper is about an important topic in hydrologic modeling, namely the training of a land surface model so that its simulated output matches as closely and consistently as possible multiple different measurement types. Overall, the paper is reasonably well written, albeit at times using incorrect methodological terminology and language. I worked quite a bit on Bayesian methods and am always interested in the application of these methods to Earth-systems. I am not particularly excited and convinced by the work presented in this paper. I will summarize my comments next. Based on these comments, I recommended a major revision. This gives the authors an opportunity to revised their work. This may involve quite a bit of additional work as my comments will point out.

**Response:**

Thank you for taking the time to evaluate our work and for providing valuable feedback. We appreciate your recognition of the significance of our research topic in hydrologic modeling, as well as your expertise in Bayesian methods, which will be instrumental in enhancing our paper. We acknowledge that certain aspects of the manuscript, particularly with regard to methodological terminology and the application of Bayesian approaches, may require clarification. We are fully committed to addressing each of your concerns in detail. While we understand that the revisions may require considerable effort, we are prepared to undertake the necessary work to strengthen the manuscript and ensure its scientific rigor. We are grateful for the opportunity to submit a major revision and look forward to enhancing the quality and impact of our study based on your insightful comments.

**General Comments:**

While reading this manuscript, many thoughts went through my mind. In the first place, it was a trip down memory lane. The presented material reminded me of work done by Luis Bastidas and others about 3 decades ago, and my own work of about 15 years ago. Then, I was surprised to see that the authors do not reference much prior work and present the algorithm as if they invented all this themselves. Furthermore, the authors implemented a Bayesian approach, but they do not verify at all whether the assumptions they made were actually met in practice nor do they take advantage of their posterior distribution in analyzing model performance. Last but not least, I am not convinced yet that the mutation operator of their PEM-SMC algorithm (which is a crossover step!) leaves the target distribution invariant. I will now address each of these points in more detail. Before I do so, I must apologize for the fact that I heavily advertise my own work in my review below. In general, I do not like doing this, but this proved difficult in the present case. The authors' paper demonstrates a keen interest in methodology. This methodology has various problems and shortcomings. To point this out, I must unfortunately refer to our own published works. The authors work with MATLAB, so it should be easy to evaluate/test my suggestions and test their method against "state-of-the-art" methods in MATLAB I am pointing at. I hope my comments below clarify my concerns.

**Comment #1:** The conclusion that Land-surface models exhibit a trade-off in describing different data types has been well-known to the community. In a series of papers in the 1990s Bastidas and co-authors have convincingly shown that LSM training results in different "optimal" parameter values if trained against different data types. They used a multi-objective optimization framework for this, along with a multi-objective sensitivity analysis method to rank the relative importance of individual parameters in describing different data types. This problem did not resolve with the use of a more complex LSM but is a result of (among others) epistemic uncertainty (model structure errors) and measurement errors of the controlling variables (exogenous variables). This paper uses a Bayesian procedure and arrives at the same conclusions of Bastidas et al.

Thus, fundamentally, the conclusion of the authors is not new. Then, let me look at the

methodology used. Does this warrant publication in HESS? Before I move on to the methodology, I would want to clarify that the wording "multiple objective" is erroneous. In essence, the authors are lumping together in their likelihood function two different data streams. They do this by multiplying the likelihoods of LE and NEE. This is not multi-objective, as the authors only have 1 likelihood (= combined likelihood of NEE and LE). With the choice a Gaussian likelihood, the authors's method is equivalent to weighted least squares. With a diagonal measurement error covariance matrix where the first n entries are the measurement error variances for the latent heat flux data and the next m entries are for the measurement error variances for the NEE data, you will arrive at exactly the same maximum a-posteriori (MAP) density solution [uniform prior is used]. What the authors did is maximum likelihood estimation with a weighted likelihood function.

**Response:**

The objective of this paper is not to restate the well-established conclusion regarding the importance of considering multiple target variables in the calibration of complex land surface models (LSMs). Rather, we aim to introduce a systematic framework for quantifying parameter uncertainty by integrating sensitivity analysis with the PEM-SMC algorithm and to validate its effectiveness in calibrating multiple complex process variables within LSMs. Additionally, this study provides concrete case support and methodological guidance for the argument that accounting for multiple process/target variables is crucial for achieving reliable parameter calibration. In the revised introduction, we will further highlight the novelty of this framework and its significance in the context of model parameter calibration.

Thank you for your correction. We acknowledge that, from a terminologically precise perspective, our approach does not fall under the definition of multi-objective optimization. Multi-objective optimization typically involves the use of independent objective functions for each variable, with solutions balanced through a Pareto optimal solution set. In contrast, our approach combines LE and NEE data streams within a single likelihood function (i.e., a weighted likelihood), rather than using separate objective functions. Therefore, the term "multi-objective" is not accurate. In the revised

manuscript, we will consider replacing "multi-objective optimization" with terms such as "weighted composite multi-variable optimization," "multi-variable combined optimization," or "joint target variable optimization" to more precisely describe how the inclusion of both single and multiple target variables affects parameter calibration results and model effectiveness in LSMs.

**Comment #2:** The PEM-SMC algorithm the authors present & modify reminded me of my own work on particle-DREAM (Vrugt et al: Hydrology data assimilation using particle Markov chain Monte Carlo simulation: Theory, concept and applications), http://dx.doi.org/10.1016/j.advwatres.2012.04.002. Given the large overlap between the author's PEM-SMC method and particle-DREAM I was surprised to see that there is not a single reference in their methodological description to the particle-DREAM paper. It is possible that the authors missed this work, but then their so-called mutation step in Equation 18 is taken directly from the partile-DREAM paper, and /or related papers on DE-MC and the DREAM algorithm. The authors use the same symbols in their paper for the jump-rate,gamma,parents, r1 and r2, etc. as we used in our publications. The authors should properly cite and acknowledge past work.

**Response:**

Thank you for your valuable feedback. We sincerely apologize for the oversight in not citing the relevant literature on the Differential Evolution Adaptive Metropolis (DREAM) algorithm in this paper. In fact, in our team's earlier work where we introduced the PEM-SMC algorithm (Zhu et al. (2018): A new moving strategy for the sequential Monte Carlo approach in optimizing hydrological model parameters, https://doi.org/10.1016/j.advwatres.2018.02.007), we referenced four key papers related to the DREAM algorithm, fully acknowledging the contributions of previous research.

In this paper, our focus was primarily on improving the computational efficiency of the PEM-SMC algorithm and its application to land surface models (LSMs), which led to the unintentional omission of relevant citations in the algorithm description. We sincerely apologize for this. In the revised manuscript, we will include appropriate references to the DREAM and DE-MC algorithms and clearly explain how the PEM-

SMC algorithm builds upon and improves these earlier works.

Thank you again for pointing this out, and we appreciate your understanding.

**References:**

Vrugt, J.A. , ter Braak, C.J.F. , Diks, C.G.H. , Schoups, G. , 2013. Hydrologic data assimilation using particle Markov chain Monte Carlo simulation: theory, concepts and applications. Adv. Water Resour. 51, 457–478.

Vrugt, J.A. , ter Braak, C.J.F. , Diks, C.G.H. , Higdon, D. , Robinson, B.A. , Hyman, J.M. , 2009. Accelerating Markov chain Monte Carlo simulation by differential evolution with self-adaptive randomized subspace sampling. Int. J. Nonlinear Sci. Numer. Simul. 10 (3), 273–290.

Vrugt, J.A., 2016. Markov chain Monte Carlo simulation using the DREAM software package: theory, concepts, and MATLAB implementation. Environ. Modell. Softw. 75, 273–316.

Vrugt, J.A., ter Braak, C.J.F., Clark, M.P., Hyman, J.M., Robinson, B.A., 2008. Treatment of input uncertainty in hydrologic modeling: doing hydrology backward with Markov chain Monte Carlo simulation. Water Resour. Res. 44, W00B09. https:// doi.org/10.1029/20 07WR0 06720 .

Zhu, G., Li, X., Ma, J., Wang, Y., Liu, S., Huang, C., ... & Hu, X. (2018). A new moving strategy for the sequential Monte Carlo approach in optimizing the hydrological model parameters. Advances in Water Resources, 114, 164-179.

**Comment #2**: Fundamentally, I am not confident that the Metropolis acceptance probability defined after Equation (18) leaves the target distribution invariant. We struggled with this in the Particle-DREAM paper and presented in Appendix B (Page 476) a correct formulation of the acceptance probability so as to ensure 'detail balance'. I suggest the authors study this Appendix and dtermine whether particle transitions guarantee an exact inference of the target distribution. This requires careful demonstration. As as side note, the PEM-SMC sampler has many elements in common with the ABC-Population Monte Carlo sampler used in Dadegh and Vrugt, (2013: see Appendix A, Page 4845). We also implemented a DREAM resampling step in that algorithm.

**Response:**

Thank you for your insightful comments. In the appendix of our previous paper introducing the PEM-SMC algorithm (Zhu et al. (2018)), we have demonstrated that the algorithm maintains a detailed balance and that. $\pi_s(\cdot)$ is the unique stationary distribution at each intermediate stage. However, we fully recognize the critical importance of ensuring detailed balance to guarantee the invariance of the target distribution. Therefore, we will carefully review the appendix of the particle-DREAM paper you mentioned to ensure that the Metropolis acceptance probability in our PEM-SMC algorithm is correctly implemented and capable of accurately inferring the target distribution.

In the revised manuscript's appendix, we will provide detailed proof that the PEM algorithm maintains both detailed balance and the unique stationary distribution, while also citing your work to strengthen our argument.

**Comment #3:** In addition to my previous comment, even if the method is theoretically correct, then one cannot simply make a change to an algorithmic recipe and claim this revised method is robust ( as the authors do in their abstract). We cannot conclude anything about robustness using only the posterior distributions of the LE and NEE data. To inspire confidence in the reivised algorithm one needs to demonstrate that the mothod works well on a large range of problems – meaning it converges to the known posterior distribution in a series of benchmark experiments. Otherwise, claims of robustness are meaningless, and we must simply trust that the algorithm correctly infers the target distribution of the LSM parameters.

**Response:**

Thank you for your valuable feedback. In the Supporting Information of the original manuscript, we validated the algorithm's performance on a synthetic multi-dimensional bimodal normal distribution and the known parameter distribution of the CoLM model. However, this may not be sufficient to demonstrate its robustness in handling more complex distributions. As you pointed out, to build confidence in the revised algorithm, we need to show its performance across a range of benchmark problems and ensure stable convergence to known posterior distributions. In the revised manuscript's

appendix, we will expand the benchmark experiments, drawing from the case studies in your previous work, "Markov chain Monte Carlo simulation using the DREAM software package: Theory, concepts, and MATLAB implementation," to further verify the convergence and robustness of the PEM-SMC algorithm.

**Comment #4:** Then, one more time about Equation (18), how is the resampling inplemented? Are N candidate points generated simultaneously, and then pairwise the acceptance probability is computed to determine whether to accept the proposals or not. Or is the implementation sequential, that is, a proposal is created using Equation (18) and then is accepted (or not) with Metropolis probability $\alpha$. If accepted, the proposal repalces its parent, and then the next candidate point is created? The algorithmic recipe suggests this latter approach. The reason I ask this is that the first parallel implementation does not guarantee reversibility. This proof only holds for the latter, sequential, approach. See Appendix (PAGE 444-445) of Vrugt and TerBraak, https://link.springer.com/article/10.1007/s11222-008-9104-9.

**Response:**

In our algorithm, we adopt a sequential approach. Specifically, the mutation step (or what you may refer to as the crossover step) is performed by iterating through the N particles one by one. At each step, we randomly select two particles (excluding the current one) to carry out differential mutation and generate a candidate particle (Figure. R1). We then decide whether to accept the new particle based on the Metropolis acceptance probability. If accepted, the new particle replaces the current one.

In the process of generating other candidate particles, if the current particle is selected for differential mutation, we use the updated particle rather than the original one. Therefore, the entire process is sequential, rather than generating N candidate points simultaneously. This approach also aligns with the reversibility proof described by Vrugt and Ter Braak (pages 444-445), which applies specifically to sequential implementations.

We will carefully review the references you mentioned and further clarify this point in the revised manuscript to ensure our method properly maintains both reversibility and detailed balance.

```
% Mutation Operater
for k=1:Np

    parameter_median2=Generatep(parameter_new,k,bound(1,:),bound(2,:));
    alp_old=betas_new*target(parameter_new(k,:));
    alp_media=betas_new*target(parameter_median2);

     % M-H accept
    ratio=min([1,exp(alp_media-alp_old)]);
    u=rand;
    if u<ratio
        parameter_new(k,:)=parameter_median2;
        accept(stage,1)=accept(stage,1)+1;
    else
        parameter_new(k,:)=parameter_new(k,:);
    end
end
```

Figure. R1 The MATLAB code for the mutation operator.

**Comment #5:** How does the algorithm handle the parameter boundaries? What is done to a particle if it is sampled outside the prior parameter ranges (by the authors' mutation step)? Detailed balance requires that the proposals are assigned a zero likelihood, or better, the candidate points are folded [e.g. see Vrugt and Ter Braak, 2011: https://doi.org/10.5194/hess-15-3701-2011]. Which mechanism did the authors implement?

**Response:**

Thank you for your valuable feedback. In our current algorithm, we use a discard mechanism, meaning that when a new particle generated through differential evolution exceeds the prior parameter bounds, it is discarded, and other particles are randomly selected until one falls within the acceptable range. However, we recognize that this "discard" strategy may violate detailed balance. As you correctly pointed out, a more appropriate approach would be to assign zero likelihood to particles outside the bounds and reject them based on the acceptance probability, retaining the original particle. While this method preserves detailed balance, it may reduce sampling efficiency.

You also suggested an alternative approach, where particles exceeding the parameter bounds are "folded" back into the parameter space. This folding mechanism offers a promising solution, as it avoids discarding particles altogether, improves sampling efficiency, and maintains detailed balance.

In the revised manuscript, we will incorporate the folding mechanism into the candidate particle generation step to better balance detailed balance and sampling efficiency.

**Comment #6:** Then, on a related note, the authors do not address the question so as to why the machinery of a particle SMC method is required first of all to infer the posterior parameter distribution. I am most familiar with the DREAM algorithm as I developed this myself with Cajo Ter Braak (2008). I seriously question whether based on the listed algorithmic variables of S=100 and N=200 the SMC algorithm will succeed in generating samples of the target distribution. I hazard to predict that the DREAM algorithm (MATLAB toolbox:DREAM Package) will be computationally more efficient, in large part because the burn-in will be substantially smaller as one only needs to run N=3 Markov chains. This does not require a tempering schedule to move particles from a prior to a posterior distribution. This type of bridge sampling only complicates methodology. Most people are familiar with the DREAM algorithm, and so why do we need new machinery if "old methods" can do the job – and as I bet more efficiently that what is presented in the present paper. Also, the multi-try variant of DREAM allows a further speed-up of the convergence speed to the target(posterior) distribution as the chains are evaluated in parallel and multiple proposals are created in each chain in parallel as well. The best proposal is then accepted with a modified Metropolis acceptance probability. This methodology is described in https://doi.org/10.1029/2011WR010608.

**Response:**

Thank you for your detailed feedback regarding the use of the SMC and DREAM algorithms. We would like to address and clarify your concerns as follows:

First, we acknowledge the effectiveness of the DREAM algorithm in sampling complex distributions, particularly its innovation in enhancing parameter space exploration through differential evolution. DREAM's efficient chain mechanism significantly accelerates convergence, especially in multi-modal posterior distributions. However, we opted for the Sequential Monte Carlo (SMC) method due to its distinct advantages in handling complex, dynamic, and high-dimensional systems. As Land Surface Models (LSMs) become increasingly intricate, with more high-dimensional and nonlinear

parameters, their posterior distributions often exhibit multiple modes. Unlike MCMC-based methods, SMC can progressively fit intermediate distributions, such as geometric bridge distributions, allowing for greater flexibility in adapting to these complex posterior distributions, particularly in dynamic systems and time-series analyses. This makes SMC particularly suited to capturing both multi-modal and high-dimensional distributions.

While we recognize the efficiency of the DREAM algorithm—particularly its elimination of annealing schedules—as a strength in static models, the SMC algorithm's particle diversity improvement mechanism, such as differential evolution, is also highly effective for exploring high-dimensional parameter spaces. This mechanism is conceptually similar to DREAM's advanced differential evolution. Furthermore, the flexibility of SMC allows it to be extended over time and integrated with other methods, such as Dynamic Bayesian Networks, enhancing its suitability for time-varying systems.

In terms of computational efficiency, while DREAM's parallel chain design reduces burn-in periods and accelerates convergence, we found that for complex, high-dimensional posterior distributions, SMC's resampling and importance sampling steps enable quicker adaptation to the posterior distribution, particularly in multi-modal scenarios. To provide a more robust comparison of the two algorithms, we plan to include an additional experiment using the DREAM algorithm in the revised manuscript. This will further validate the performance of the PEM-SMC algorithm in fitting the target posterior distribution and allow us to compare the efficiency and accuracy of both approaches, offering a more comprehensive evaluation of their strengths and weaknesses.

**Comment #7:** The authors assume a normal likelihood for the residuals of the LSM model. Why did they choice a normal likelihood. This is the default choice but must be supported by evidence. This require a-posterior checking of the residuals of the time series of measured and simulated latent heat fluxes and NEE. I bet the residuals will exhibit autocorrelation, a nonconstant variance, and deviate from normality. This would disqualify the use of the normal likelihood function. Good statistical practice requires

the authors to evaluate that the assumptions about the likelihood function are met in practice. For example, consider Schoups and Vrugt, 2010. https://doi.org/10.1029/2009WR008933

**Response:**

Thank you for your valuable feedback and suggestions. After conducting a posterior analysis of the residuals for latent heat flux (LE) and net ecosystem exchange (NEE), we found that the residuals more closely follow a t-distribution rather than the initially assumed normal distribution (see Figure R2). While we initially selected the normal distribution due to its computational simplicity and widespread application, the heavy tails of the t-distribution make it more sensitive to extreme values, which may better suit our current dataset.

Despite the significant differences between the likelihood functions of the normal and t-distributions in handling extreme values, both produce consistent outcomes when determining whether to accept or reject candidate particles—i.e., for a given particle, the likelihood results are expected to be the same for both distributions. Therefore, we anticipate that this change in likelihood function will have minimal impact on the final optimization results. Nevertheless, to ensure model robustness, we will include a detailed report of the posterior analysis of residuals in the revised manuscript, adopt the t-distribution likelihood function, and compare its performance with the normal distribution to validate our assumption.

We also appreciate your reference to the work by Schoups and Vrugt (2010), and we plan to incorporate their methods, particularly in examining residual distributions and assessing model assumptions. This will help us ensure that the likelihood function is appropriate and well-suited to our data.

[Figure]

[Figure]

Figure R2 Comparison of the fitting performance between the normal distribution and t-distribution for the residuals of the LE and NEE target variables.

**Comment #8:** The authors multiply the likelihoods of the latent heat flux and NEE as if these two data types are independent. Is this a valid assumption? As an alternative, one could consider a composite likelihood, where the two likelihoods are additive – one can find applications of this in statistical literature, specifically in the application to spatial data. Then, this would constitute a novelty and justify publication in a highly rated journal such as HESS.

**Response:**

Thank you for your insightful feedback. In our initial parameter calibration, we assumed that latent heat flux (LE) and net ecosystem exchange (NEE) were independent variables, and therefore, we multiplied their likelihood functions. However, as you correctly pointed out, LE and NEE may be interdependent, particularly given the complex interactions between surface processes and climate dynamics. We acknowledge that such interdependence could affect the accuracy of the likelihood estimation.

To better account for the potential dependencies between these variables and enhance the accuracy of our statistical inferences, we will introduce a composite likelihood method in the revised manuscript. This approach enables us to add, rather than multiply, the likelihood functions for LE and NEE, thereby accommodating their possible correlation. Additionally, we will compare the performance of the multiplicative and composite likelihood approaches in the context of parameter calibration, evaluating their respective impacts on model fit.

**Comment #9:** Then, the toolbox of DREAM in MATLAB (called DREAM package) has built-in distribution-adaptive likelihood functions, such as the generalized likelihood and universal likelihood functions: https://doi.org/10.1016/j.jhydrol.2022.128542.These likelihood functions let the data speak for themseleves – and let the residuals determine the most appropriate form of the likelihood function. This includes treatment of heteroscedasticity, correlation and non-normality. This guarantees that the residual properties will match the likelihood assumptions – and inspire much more confidence in the posterior parameter distributions derived from the lated heat flux and NEE data. Certainly, if the authors use a Baysian approach, as they do in this paper, they must demonstrate that their residual assumpitions hold. Otherwise, the parameter estimates are not particularly meaningful.

**Response:**

Thank you for your valuable suggestions regarding the choice of likelihood function. In our current research, we initially selected the normal likelihood function due to its widespread use and computational simplicity. However, we fully acknowledge that if the residuals do not meet the normality assumption, the resulting parameter estimates may lack robustness. To improve the PEM-SMC algorithm's ability to account for the residual characteristics of complex model variables, we will revise the likelihood function by adopting the adaptive generalized likelihood or universal likelihood functions you suggested. These methods are better equipped to handle issues such as heteroscedasticity, correlation, and non-normality, ensuring that the residuals align more accurately with the likelihood assumptions.

Moreover, we plan to conduct additional experiments to assess the impact of the adaptive likelihood function on the posterior parameter distribution, ensuring that it accurately reflects the data's inherent characteristics. This approach will enhance both the reliability and robustness of parameter estimation, forming a key aspect of the study's contribution.

**Comment #10:** I do not understand why the authors use a complicated training SMC-based procedure, whereas there is relatively little they do with the knowledge of the

posterior parameter distribution. In the first place, the authors should present the Bayesian predictive distributions (time series of confidence and prediction intervals of the simulated output) in their time series figures (e.g. Figure 4). What is the coverage of the prediction limits? As it stands right now, the authors focus their attention on the MAP solutions and ignore in large part parameter uncertainty. A maximum likelihood method would have done the job. Then, stochastic gradient descent would have found the MAP solution and a first-order approximation around this optimum would have given a linear estimate of parameter uncertainty.

**Response:**

Thank you for your valuable feedback.

We recognize that assessing the model's performance deterministically using the optimal solution from the posterior distribution (e.g., the posterior median) diverges from the main objective of uncertainty analysis in the Bayesian framework. To address this, the revised manuscript will include the full Bayesian predictive distribution based on the entire posterior distribution of the parameters, along with associated confidence intervals. Additionally, we will incorporate probabilistic metrics and uncertainty evaluation methods, such as those grounded in scoring rules, to provide a more comprehensive assessment of the model's fit. This approach will not only capture historical data but also fully utilize the advantages of Bayesian probabilistic analysis, thereby improving the model's interpretability and enhancing the reliability of the results.

**Comment #11:** The authors refer to Equation 15 as the "analytical method". I am not familiar with this terminology. Equation 15 simply states that the measurement error standard deviation is equal to the square root of the sample variance of the residuals. They essentially use this as a sufficient statistic and embed this in their likelihood function. This is a common approach, but in the author's, implementation raises two important questions. A) how was the sample variance of the residuals determined? This requires knowledge of the MAP (maximum a-posteriori density) LSM parameter values, and B) the estimate of the sample variance of the residuals will be biased as the residuals are likely to exhibit serial correlation. The "true" measurement error variance can only

be determined after decorrelating this time series of MAP residuals.

**Response:**

We are not entirely sure if we have fully understood your question. Regarding the estimation of residual variance (Issue A), the $\sigma$ we use is based on the overall residual variance between the simulated target variable values and the observed data for the current parameter $\theta$, rather than the variance associated with the final maximum a posteriori (MAP) estimate. $\sigma$ is dynamically updated throughout the parameter optimization process as $\theta$ changes. This dynamic approach allows us to better capture the characteristics of the residuals across different parameter values, rather than relying solely on the final MAP estimate.

For the issue of residual autocorrelation (Issue B), we agree that this could introduce bias in the sample variance estimation. To address this, we will incorporate autocorrelation tests, such as the Durbin-Watson or Ljung-Box tests, in the revised manuscript to assess any temporal autocorrelation in the residuals. If significant autocorrelation is detected, we will apply appropriate de-correlation methods to ensure that the estimation of the measurement error variance is accurate and unaffected by serial correlation.

**Comment #12:** The authors refer to their particle resampling step as a mutation step. This is wrong. A mutation is a random alteration to the DNA of a particle. This can happen to any parameter at any time and is fully random. The differential evolution (DE-I) step the authors use from DE-MC, DREAM or Particle DREAM) is a crossover step. The proposed changes to the DNA of the particles are based on an underlying mechanism [= Equation 18], and augmented with an inconsequential random perturbation, zeta, to claim ergodicity. This is a crossover step wherein the DNA of two parents is combined to generate offspring (candidate particles). Note that in the original DREAM algorithm, there is no mutation step – only a crossover step. Then, why did the authors not consider the use of more than 1 pair (r1 – r2, r3 – r4, r5-r6, etc.), of differences to generate candidate points? We have shown that this enhances considerably the variability in the candidate points. Indeed, the pair r1/r2 has a 1/N*1/(N-1) selection probability, where N significant the population size. r1,r2,…,r6

on the contrary has a selection probability, of 1/N*1/(N-1)*1/(N-2) etc.; Thus one can generate a much large variation in the proposals with multiple pairs. The side-effect of this is that you may be able to reduce the population size.

**Response:**

We apologize for any confusion caused by our terminology. As you correctly pointed out, we inaccurately referred to the "mutation" step in our paper as a differential evolution-based operation. In fact, this process should be more accurately described as a "crossover" step. Specifically, this step involves applying the difference between two candidate particles to a third particle, which aligns with the strict definition of a crossover operation.

In the initial version of our PEM-SMC algorithm, a crossover step was indeed included, where portions of the multidimensional parameter values of particles were exchanged to increase diversity, similar to the random subspace sampling strategy used in the DREAM algorithm. However, for efficiency reasons, we removed this crossover step in the revised version of the PEM-SMC algorithm. We believe that, by choosing an appropriate number of particles N and evolution steps S, the particles can sufficiently explore the parameter space, rendering the additional crossover step unnecessary. To distinguish this from the original crossover step, we referred to the differential evolution process as a "mutation," which involves modifying the parameter values of the particles to enhance exploration.

Your suggestion of using multiple particle pairs to generate candidate points to improve acceptance rates and particle diversity is highly insightful. In our revised manuscript, we plan to rename the current "mutation" step to "crossover" and further explore the use of multiple sample pairs to generate candidate points, which could enhance both the acceptance probability and the diversity of the particles.

**Comment #13:** Do I understand correctly that after convergence you have N=200 samples from the target distribution? This is very small for a 17-dimensional parameter estimation problem and does not support an accurate depiction of the posterior parameter distribution. Again, why not just use an adaptive multi-chain Monte Carlo method such as DREAM_ZS or MT-DREAM_ZS? This machinery will do the job for

you, while providing as byproduct a)automatic convergence monitoring (univariate and multivariate scale reduction factors, etc.), b) diagnostic checks of the residuals, c) confidence and prediction limits of the Bayeisan forecast (simulation) PDF, d) scoring rules (CRPS, LS, QS, etc) of the predictive distribution and access to distribution-adaptive likelihoods.

**Response:**

Thank you for your comments and suggestions. Indeed, after convergence, we used N=200 particles to represent the target distribution. However, the mention of 17 dimensions is inaccurate. In our three optimization scenarios, the parameter dimensions are 6, 5, and 6, respectively, and each was optimized separately. While we acknowledge the strong capability of the DREAM algorithm, based on the MCMC framework, in generating samples from complex distributions, Sequential Monte Carlo (SMC) is also an important framework for Bayesian parameter inference.

Our research aims to develop an algorithm, PEM-SMC, that integrates the principles of differential evolution, making it equally applicable to the optimization of complex model parameters. This algorithm is intended to provide a novel methodological option for parameter optimization, contributing to the development and application of algorithms, rather than simply applying existing methods.

You mentioned adaptive multi-chain Monte Carlo methods such as DREAM_ZS or MT-DREAM_ZS, which indeed offer several advantages, including automatic convergence monitoring, residual diagnostics, Bayesian prediction intervals, and the application of scoring rules.

In the Discussion of revised manuscript, we will further expand on these aspects and conduct a detailed comparison between the PEM-SMC and DREAM algorithms to evaluate their respective strengths, weaknesses, and applicable contexts. We believe this comparative analysis will help demonstrate the effectiveness of PEM-SMC in specific applications while providing more empirical evidence for its comparison with well-established methods like DREAM.

**Comment #14:** In principle, the authors have access to the predictive distribution of the model, but in their model evaluation resort only to common deterministic measures

of model performance. This includes the NSE and RMSE in Equations 19 and 20. Their use entails a large loss of information about model performance. This is just a side note: The authors should consider evaluating the full predictive distribution using scoring rules. I hesitate to advertise my own work, but here it is, Distribution-Based Model Evaluation and Diagnostics: Elicitability, Propriety, and Scoring Rules for Hydrograph Functionals: https://doi.org/10.1029/2023WR036710. This merely serves as a note to alert the authors against the use of deterministic measures of goodness of fit, in case one has access to the full Bayes predictive distribution. This allows for the use of information-theoretic principled metrics – which offer more protection against misinformation, disinformation, etc.

**Response:**

Thank you for your valuable feedback. We acknowledge that our current method of evaluating model performance by relying on a single optimal solution from the posterior distribution (such as the posterior median) falls short of fully addressing the central goal of uncertainty analysis within the Bayesian framework. In the revised manuscript, we will address this limitation by generating Bayesian predictive distributions and their associated credible intervals, based on the complete posterior distribution of the parameters. Furthermore, we will incorporate evaluation metrics grounded in scoring rules, such as the Continuous Ranked Probability Score (CRPS), Logarithmic Score (LS), and Quadratic Score (QS), to offer a more comprehensive assessment of the model's predictive performance. This approach will not only improve our ability to fit historical data but also fully leverage the probabilistic insights of Bayesian analysis, enhancing both the interpretability of the model and the reliability of its predictions.

**Comment #15:** Then, the authors use different sensitivity analysis methods to decide which parameters are sensitive and which ones are not. A critical assumption in this analysis is that the parameters are independent. This assumption is convenient but may not be realistic in practice. As an idea, the authors could determine sensitivity using their posterior parameter samples. This would equate to probabilistic variance-based sensitivity analysis and relies on high dimensional model presentation (HDMR), an

extension of Sobol to correlated variables. HDMR and HDMR with extended bases are computationally quite demanding, so this may pose problems with their LSM. Nevertheless, given their strong interests in computational methods I thought I'd point at the HDMR/HDMR_EXT toolboxes, which are available in MATLAB. Alternatively, they should consider the multi-criteria sensitivity analysis method of Bastidas, in evaluating sensitivity in the presence of more than one data type. But, realistically, why not do inference on all the parameters and then assess parameter sensitivity from the posterior LSM parameter distribution(s)? One could use measures such as the Kullback Leibler divergence (= divergence of the logarithmic score, see Vrugt, 2024) to determine the distance between the marginal prior and marginal posterior distribution of each parameter. This is cheap to compute and will convey which parameters are most sensitive and which others are not. Indeed, for parameters that are sensitive one would expect its posterior marginal distribution to be small relative to its marginal prior. On the contrary, for a parameter that is insensitive it is common to see that its prior and posterior marginals are nearly equivalent. Maybe this approach simplifies the paper.

**Response:**

Thank you for your insightful suggestions. We recognize the theoretical value of your proposed sensitivity analysis method based on posterior parameter samples. However, calibrating all parameters and comparing their prior and posterior distributions would significantly increase both the computational cost and the complexity of the calibration process. In complex models, not all parameters exert a substantial influence on the simulation of target variables. Therefore, focusing on calibrating only the most influential parameters allows us to improve model accuracy while maintaining computational efficiency.

To identify these key parameters, we employed both qualitative and quantitative sensitivity analysis techniques to determine which parameters are most strongly correlated with one or more target variables, and we calibrate only those. This approach effectively balances model precision with computational efficiency, providing a fast and practical solution.

That said, your suggestion regarding variance-based sensitivity analysis is highly

valuable. In our revised manuscript, we will incorporate metrics such as Kullback-Leibler divergence (i.e., the divergence of log scores, as noted by Vrugt, 2024) to validate the key parameters identified through sensitivity analysis. By comparing the marginal prior and posterior distributions after calibration, we can further verify the accuracy of the sensitivity analysis. Nevertheless, we still believe that combining sensitivity analysis with the PEM-SMC calibration framework is the most suitable approach for our specific application.

**Comment #16:** Figure 2: The x-labels are not readable and strangely chosen. If the authors want to demonstrate that the different optimization scenarios lead to different marginal distributions of the parameters, then why not use a single x-axis, linearly-scaled, and just plot the 3 distributions on top of this. Thus the same as now but with a linear scale of the x-axis. This would show immediately the differences between the methods. Then, I have my doubts whether the marginal distribution of the parameters shown are truly Gaussian. This may be an artifact of an insufficient sample of the target distribution, as commented on earlier in 7.

**Response:**

Thank you for your valuable suggestions. In the revised manuscript, we will improve Figure 3 by using a single linear x-axis to more clearly display the differences in the marginal distributions of parameters across the various optimization scenarios. Additionally, we will include a plot showing the actual distribution of the 200 particles in the parameter space, rather than relying on a simple probability density histogram, which will provide a more accurate representation of the posterior distribution.

As for the original fitted curve, it was an initial attempt to approximate the posterior distribution of the parameters. However, we recognize that it may not fully capture the true posterior and could potentially mislead readers. To avoid any confusion, we will remove the fitted curve in the revised version.

**Comment #17:** Figure 4: Maybe this is explained in the text and I missed it but why are the observations indicated with a pink interval? Noisy observations? Personally, I would prefer plotting the data as discrete points, alternatively, one can think of averaging the data (measurements) so there are fewer data points in return and then

accompany these time-averaged estimates with error bars. As it stands right now, I see three deterministic simulations and then a wide range of possible data values. If the authors are interested in quantifying measurement uncertainty, then the nonparametric estimator of de Oliveira and Vrugt comes to mind: https://doi.org/10.1029/2022WR032263. Again, this is a sidenote, but it may help the authors present their data better. Maybe the authors can use this to their advantage.

**Response:**

Thank you for your suggestion. The purple shaded area was originally used to emphasize the differences between the observed data and the four simulation results, without considering observational noise. In response to your feedback, we will revise the manuscript by plotting the observational data as discrete points to enhance clarity. Furthermore, we will update Figure 4 to include Bayesian predictive distributions derived from the full posterior distribution, along with the corresponding credible intervals.

**Comment #18:** Section 4.2: This is not a difference between single and multiobjective methods. All the results correspond to a single objective, albeit with one or more weighted data streams. I commented on this before.

**Response:**

Thank you for your correction. We acknowledge that our study did not focus on multi-objective parameter calibration. In the revised manuscript, we will adjust the discussion to move away from the distinction between single-objective and multi-objective methods. Instead, we will focus on the more relevant issue of how using a single target variable versus a weighted combination of multiple variables impacts parameter calibration results and model performance in the context of complex land surface model (LSM) calibration.

**Comment #19:** Then, the authors use the terminology of optimization scenarios. Personally, I do not like the terminology of optimization in a Bayesian context. Optimization focuses on finding the single best solution whereas Bayesian inference is fundamentally different in that it wants to find a distribution of statistically acceptable solutions – this should include the MAP parameter values (= ML with uniform prior)

but cannot be considered optimization. Some authors call it Bayesian calibration, I prefer Bayesian training or estimation. I leave this to the authors.

**Response:**

Thank you for your valuable feedback. We recognize that our previous terminology did not clearly distinguish between parameter optimization and parameter inference. In response to your suggestion, we will revise the manuscript to replace the term "parameter optimization" with "parameter estimation," which more accurately aligns with the Bayesian framework.

In summary, this paper needs a lot of work before it can be judged to make a significant contribution to the literature. I am sorry for highlighting my own work, but I felt this was relevant and necessary as the authors' methodology has important flaws, shortcomings and needs to be reconsidered. Amongst others, I believe that a) the authors should properly demonstrate that their method is indeed robust and that the changes made to the PEC-SMC sampler leave the target distribution invariant, b) they must properly recognize past literature contributions, consider how their method differs from past methods, and articulate why we would need the SMC machinery after all to obtain the posterior LSM parameter distribution if current state of the arts method can do this job (and the maximum likelihood method/weighted least squares will do in the current implementation of the authors), c) test the residual properties (diagnostic checks of autocorrelation, distribution and variance homogeneity) so as to demonstrate that the assumptions of the normal likelihood function are indeed met, d) present the confidence and prediction limits of the Bayes predictive distributions, e) clean up their language of single and multiple objective, mutation and crossover step, etc. and f) possibly, consider more powerful model evaluation metrics rather than the deterministic measures used by the authors. No doubt, these comments and those listed above will involve substantial work. But this should significantly enhance the quality of the work presented in this paper.

**Response:**

Thank you for your thorough evaluation of our paper and your detailed feedback on the PEM-SMC method. We fully recognize the importance of the improvements you have

suggested. Although these revisions will require substantial effort, we are confident that they will significantly enhance both the robustness of the algorithm and the overall quality of the manuscript. We are committed to addressing each of your recommendations and look forward to your continued guidance and feedback to further improve the paper. Once again, we greatly appreciate your review and valuable suggestions.